# Measure Theory of Conditionally Independent Random Function Evaluation

## Abstract

In sequential design strategies, common in geostatistics and Bayesian optimization, the selection of a new observation point $X_{n+1}$ of a random function $\mathbf{f}$ is informed by past data, captured by the filtration $\mathcal{F}_n = \sigma(\mathbf{f}(X_0), \ldots, \mathbf{f}(X_n))$. The random nature of $X_{n+1}$ introduces measure-theoretic subtleties in deriving the conditional distribution $\mathbb{P}(\mathbf{f}(X_{n+1}) \in A \mid \mathcal{F}_n)$. Practitioners often resort to a heuristic: treating $X_0, \ldots, X_{n+1}$ as fixed parameters within the conditional probability calculation. This paper investigates the mathematical validity of this widespread practice. We construct a counterexample to prove that this approach is, in general, incorrect. We also establish our central positive result: for continuous Gaussian random functions and their canonical conditional distribution, the heuristic is sound. This provides a rigorous justification for a foundational technique in Bayesian optimization and spatial statistics. We further extend our analysis to include settings with noisy evaluations and to cases where $X_{n+1}$ is not adapted to $\mathcal{F}_n$ but is conditionally independent of $\mathbf{f}$ given the filtration.

## 1 Introduction

Researchers have been confronted with the challenge of optimizing an unknown function across numerous disciplines. The Bayesian approach addresses this challenge by modelling the unknown function as a random draw from a prior distribution over functions. This probabilistic framework ensures the conditional distribution of a *random function*[1] $\mathbf{f} = (\mathbf{f}(x))_{x \in \mathbb{X}}$ given a set of function evaluations $\mathbf{f}(x_1), \ldots, \mathbf{f}(x_n)$ is well-defined. This conditional distribution can then be used to choose the next evaluation point. This approach to optimization has independently emerged across multiple research domains, each developing its own terminology for very similar methods. The most prominent is perhaps *Bayesian optimization* (Kushner, 1964; Jones et al., 1998; Frazier, 2018; Garnett, 2023), which is best know in machine learning for its effectiveness at hyperparameter tuning. However similar techniques were already developed earlier in *geostatistics* known as "Kriging" (Krige, 1951; Matheron, 1963; Stein, 1999).

In this paper we highlight and address measure-theoretic challenges that must be overcome for a rigorous mathematical treatment of random function optimization. In particular, we show how non-rigorous but intuitive claims about the conditional distributions of the objective function $\mathbf{f}$ can be made rigorous. In our main result we formalize the heuristic that evaluation locations $X_n$, which are measurable with respect to $\mathcal{F}_{n-1} = \sigma(\mathbf{f}(X_0), \ldots, \mathbf{f}(X_{n-1}))$ ("previsible"), may be treated as if they are deterministic during the calculation of the conditional distribution

$$\mathbb{P}(\mathbf{f}(X_n) \in \cdot \mid \mathbf{f}(X_0), \ldots, \mathbf{f}(X_{n-1})).$$

If there is a formula for such a conditional distribution **for all** deterministic $X_k$, then we call this formula a *joint* kernel (Definition 2.1). If this formula is furthermore valid **for all** previsible random $X_k$, then we call it "plug-in consistent" (PIC) (see Definition 2.3 and 2.8). Standard disintegration results can only provide

---

[1]While used synonymously, we avoid the more common term 'stochastic process' which invokes the notion of a one-dimensional index representing 'time' and a filtration associated to this time. The domain $\mathbb{X}$ is generally unordered, e.g. $\mathbb{X} = \mathbb{R}^d$, and the filtration we consider naturally arises from the sequence of evaluations of this random function $\mathbf{f}$.

results for almost all deterministic $X_k$ with respect to a single distribution over $X_k$. One may therefore call our results "strong disintegration results" (see Remark 2.2). In Counterexample 2.13 we show that the PIC property is not guaranteed if the assumptions of our main results are violated.

*Remark* 1.1 (Measurability of random evaluations). A priori, even the measurability of $\mathbf{f}(X)$ – its very existence as a random variable – is not guaranteed. However, a measurable evaluation map $e(f, x) := f(x)$ would provide this guarantee and we show that this requirement is satisfied in considerable generality: If $\mathbf{f}$ is a *continuous* random function with locally compact, separable, metrizable domain $\mathbb{X}$ and Polish codomain $\mathbb{Y}$, then the evaluation map $e$ is continuous, and hence measurable (see Theorem B.1). This covers essentially all continuous applications of interest (e.g. $\mathbb{X} \subseteq \mathbb{R}^d$ and $\mathbb{Y} = \mathbb{R}^n$). The difficulty lies in the fact that the evaluation map is only well known to be continuous with respect to the compact-open topology, which is different from the product topology used to generate the Borel $\sigma$-algebra for random functions (Remark B.2).

**Outline**  In Section 2 we present the theory, first for a simplified case in Section 2.1, then for the general case in Section 2.2. For the case where the assumptions of these results are violated we present a counterexample in Section 2.3. In Section 3 we present applications of our results. Section 4 contains the proofs for the results from Section 2. In Section B we address the issues outlined in Remark 1.1. While these results are likely to be known, we could not find them anywhere in this generality.

## 2 Measure theory of random function evaluation

Throughout the paper we assume there exists an underlying probability space $(\Omega, \mathcal{A}, \mathbb{P})$.

### 2.1 Simplified case: Dependent evaluation

Let $\mathbf{f} = (\mathbf{f}(x))_{x \in \mathbb{X}}$ be a random continuous function with locally compact, separable metrizable domain $\mathbb{X}$ and Polish codomain $\mathbb{Y}$. That is we assume $\mathbf{f}$ is a random variable in the space of continuous functions $C(\mathbb{X}, \mathbb{Y})$. In this section we consider the simplified case

$$\mathbb{P}(\mathbf{f}(X) \in A \mid \mathcal{F}) \tag{1}$$

with only a single random evaluation point $X$ in $\mathbb{X}$ that is measurable with respect to a sub-$\sigma$-algebra $\mathcal{F}$ of $\mathcal{A}$ and sets $A$ in $\mathcal{B}(\mathbb{Y})$. Here $\mathcal{B}(\mathbb{Y})$ denotes the Borel $\sigma$-algebra on $\mathbb{Y}$. Assume we have access to a formula to calculate (1) for any deterministic $X$. Specifically, assume we have a collection $(\kappa_x)_{x \in \mathbb{X}}$ of regular conditional distributions such that for all $x \in \mathbb{X}$, $A \in \mathcal{B}(\mathbb{Y})$ and for $\mathbb{P}$-almost all $\omega \in \Omega$,[2]

$$\mathbb{P}(\mathbf{f}(x) \in A \mid \mathcal{F})(\omega) = \kappa_x(\omega; A).$$

Corollary 2.5 then establishes sufficient conditions on the collection $(\kappa_x)_{x \in \mathbb{X}}$ to ensure that **for all** $\mathcal{F}$-measurable $X$, all $A \in \mathcal{B}(\mathbb{Y})$ and for $\mathbb{P}$-almost all $\omega \in \Omega$,[2]

$$\mathbb{P}(\mathbf{f}(X) \in A \mid \mathcal{F})(\omega) = \kappa_{X(\omega)}(\omega; A).$$

This means that random $X$ may be treated like deterministic $x \in \mathbb{X}$ in the "formula" $\kappa_\bullet(\omega; A)$ for $\mathbb{P}(\mathbf{f}(\bullet) \in A \mid \mathcal{F})(\omega)$.

Above it is implicitly assumed that the function $\kappa_{X(\cdot)}(\cdot; A)$ is measurable and therefore a well-defined random variable. To guarantee this, we require that $(\omega, x) \mapsto \kappa_x(\omega; A)$ is a measurable mapping. This is not guaranteed for arbitrary collections of regular conditional distributions and warrants the following definition. It represents the first restriction on the collection $(\kappa_x)_{x \in \mathbb{X}}$.

---

[2] Recall that $\mathbb{P}(\mathbf{f}(x) \in A \mid \mathcal{F})$ is only defined as an $L^1$-equivalence class of random variables. For an equivalence class $[Z]$ in $L^1$ it is standard to say, "$[Z](\omega) = Y(\omega)$ holds for almost all $\omega$", if for all $Z \in [Z]$ there exists a null set $N_Z$ such that $Z(\omega) = Y(\omega)$ for all $\omega \in N_Z^{\complement}$. Observe that the null set $N_Z$ depends on $Z$. For $\mathbb{P}(\mathbf{f}(x) \in A \mid \mathcal{F})$ this null set *must* therefore be allowed to depend on $x$ and $A$.

**Definition 2.1** (Joint probability kernels). Let $(\Omega, \mathcal{A})$, $(\mathbb{X}, \mathcal{X})$ and $(\mathbb{Y}, \mathcal{Y})$ be measurable spaces. The function $\kappa \colon (\Omega \times \mathbb{X}) \times \mathcal{Y} \to [0, \infty]$ is called a *joint kernel* for the collection $(\kappa_x)_{x \in \mathbb{X}}$ of probability kernels $\kappa_x \colon \Omega \times \mathcal{Y} \to [0, \infty]$, if

1. for all $x \in \mathbb{X}$
$$\kappa(\omega, x; A) = \kappa_x(\omega; A) \qquad \forall \omega \in \Omega, A \in \mathcal{Y},$$

2. the mapping $(\omega, x) \mapsto \kappa(\omega, x; A)$ is measurable for all $A \in \mathcal{Y}$, such that the function $\kappa$ becomes a probability kernel.

A joint kernel for a collection of regular conditional distributions is called a *joint conditional distribution*.

*Remark* 2.2 (Strong disintegration). A joint kernel is simply a kernel on the product space $\Omega \times \mathbb{X}$ with target $(\mathbb{Y}, \mathcal{Y})$. This does not mean that standard disintegration results yield such a kernel. This is because a regular conditional distribution requires a measure. On $\Omega$ this is given by $\mathbb{P}$. But the product space $\Omega \times \mathbb{X}$ is typically not equipped with a measure. If we take a fixed random variable $X$ in $\mathbb{X}$, measurable with respect to $\mathcal{F}$, then standard disintegration results imply the existence of a regular conditional probability distribution

$$\mathbb{P}(\mathbf{f}(X) \in A \mid \mathcal{F})(\omega) = \kappa(\omega; A).$$

But the kernel $\kappa$ does not take $X$ as input and is only valid for this particular random variable $X$. Similarly, we could equip the product space $\Omega \times \mathbb{X}$ with a measure $\mathbb{P}' = \mathbb{P} \otimes \nu$, where $\nu$ is a measure on $(\mathbb{X}, \mathcal{X})$. Thus $X(\omega, x) \coloneqq x$ is a random variable with respect to $\mathbb{P}'$ and $\mathbb{P}_X = \nu$. Standard disintegration results would then imply, that for all $\mathcal{F} \otimes \mathcal{X}$-measurable $B$ we have

$$\mathbb{P}'(\{\mathbf{f}(X) \in A\} \cap B) = \int_B \kappa(\omega, x; A) \mathbb{P}(d\omega) \nu(dx).$$

Or in other words,

$$\mathbb{P}'(\mathbf{f}(X) \in A \mid \mathcal{F}, X = x)(\omega) = \kappa(\omega, x; A)$$

for almost all $\omega$. However, this kernel $\kappa$ would depend on $\nu$ and would only be valid for $\nu$-almost all $x \in \mathbb{X}$. Moreover we constructed $X$ independent from $\mathcal{F}$, which required us to add $X = x$ to the conditional part. In contrast, a "joint conditional distribution" is valid **for all** $x \in \mathbb{X}$. And the PIC property we will define in the following essentially means that $\kappa$ is a regular conditional distribution **for all** distributions $\nu = \mathbb{P}_X$. One may therefore call these results "strong disintegration" results.

These stronger properties are crucial for applications. The explicit relation of the joint conditional distribution to collections of conditional distributions is crucial to calculate Bayesian optimizers (Example 3.1 and 3.5). The uniform validity over distributions $\mathbb{P}_X$ of $X$ is important for Examples 3.3 and 3.6.

Proposition 2.4 establishes the existence and uniqueness of a "plug-in consistent" (PIC) joint conditional distribution, where a PIC joint kernel also admits random input. This proposition also generalizes the setting of Equation (1) to an additional random variable $Z$, which will be a helpful tool for our main result.

**Definition 2.3** (Plug-in consistent – PIC). Let $Z$ be a random variable in the standard Borel space $(E, \mathcal{B}(E))$ and $\kappa(\omega, x; B)$ a joint conditional distribution for $\mathbb{P}((Z, \mathbf{f}(x)) \in B \mid \mathcal{F})$ indexed by $x \in \mathbb{X}$, where $(\mathbb{X}, \mathcal{X})$ is a measurable space. Then $\kappa$ is called *plug-in consistent* (PIC), if **for all $\mathcal{F}$ measurable random variables** $X$ in $\mathbb{X}$, all $B \in \mathcal{B}(E) \otimes \mathcal{B}(\mathbb{Y})$ and for $\mathbb{P}$-almost all $\omega \in \Omega$,[2]

$$\mathbb{P}\big((Z, \mathbf{f}(X)) \in B \mid \mathcal{F}\big)(\omega) = \kappa(\omega, X(\omega); B). \tag{2}$$

**Proposition 2.4** (Existence and properties of PIC conditional distributions). *There exists a PIC joint conditional distribution $\kappa$ for $(Z, \mathbf{f}(x))$ given $\mathcal{F}$ as defined in Definition 2.3.*

*This PIC kernel can be chosen such that $x \mapsto \kappa(\omega, x; \cdot)$ is continuous with respect to the weak topology on the space of measures for all $\omega \in \Omega$. Let $\tilde{\kappa}$ be another joint conditional distribution for $(Z, \mathbf{f}(x))$ given $\mathcal{F}$, which is continuous for almost all $\omega$ in this sense. Then there exists a null set $N$ such that for all $\omega \in N^{\complement}$, all $x \in \mathbb{X}$ and all borel sets $B \in \mathcal{B}(E) \otimes \mathcal{B}(\mathbb{Y})$*

$$\kappa(\omega, x; B) = \tilde{\kappa}(\omega, x; B).$$

*In particular, $\tilde{\kappa}$ is also PIC. Finally, let $g \colon E \times \mathbb{Y} \to \mathbb{R}$ be a continuous function with $\mathbb{E}\big[\sup_{x \in \mathbb{X}} |g(Z, \mathbf{f}(x))| \mid \mathcal{F}\big] < \infty$ almost surely, then*

$$\mathbf{g}(x) := \int g(z, y)\, \kappa(\cdot, x; dz \otimes dy) \stackrel{a.s.}{=} \mathbb{E}[g(Z, \mathbf{f}(x)) \mid \mathcal{F}]$$

*is almost surely continuous in $x$.*

Corollary 2.5 provides sufficient conditions to ensure collections of conditional distributions $(\kappa_x)_{x \in \mathbb{X}}$ form a joint kernel and are PIC.

**Corollary 2.5** (Random dependent evaluation). *Let $(\kappa_x)_{x \in \mathbb{X}}$ be a collection of regular conditional distributions for $\mathbf{f}(x)$ given $\mathcal{F}$, where the random function $\mathbf{f}$ is continuous with locally compact, separable domain $\mathbb{X}$ (e.g. $\mathbb{R}^d$) and Polish codomain $\mathbb{Y}$ (e.g. $\mathbb{R}^n$). Then **for all** $\mathcal{F}$-measurable $X$ and all $A \in \mathcal{B}(\mathbb{Y})$ we have*

$$\mathbb{P}(\mathbf{f}(X) \in A \mid \mathcal{F})(\omega) = \kappa_{X(\omega)}(\omega; A) \tag{3}$$

*for almost all $\omega$, if*

  (i) *the map $(\omega, x) \mapsto \kappa_x(\omega; A)$ is **measurable** for all $A \in \mathcal{B}(\mathbb{Y})$,*
  (ii) *$x \mapsto \kappa_x(\omega; \cdot)$ is **continuous** in the weak topology for almost all $\omega$.*

*Proof.* (i) ensures that $\tilde{\kappa}(\omega, x; A) := \kappa_x(\omega; A)$ is a joint kernel and (ii) ensures that $\tilde{\kappa}$ is PIC as it satisfies the requirements in Proposition 2.4. $\qquad\square$

*Remark* 2.6 (Tightness of the result). Counterexample 2.13 shows that some continuity in $x$ is necessary.

## 2.2 General case: Conditionally independent evaluation locations

In the introduction we motivated previsible sequences $(X_n)_{n \in \mathbb{N}}$ with respect to the filtration $\mathcal{F}_n = \sigma(\mathbf{f}(X_0), \dots, \mathbf{f}(X_n))$. To get to our main result we generalize this previsible setting to conditionally independent evaluation points, admit noisy function evaluation and also admit starting information.

**Conditional independence**   Sometimes $X_{n+1}$ is not previsible itself, but sampled from a previsible distribution. That is, a distribution constructed from previously seen evaluations (e.g. Thompson sampling (Thompson, 1933)). In this case, $X_{n+1}$ is not previsible, but independent from $\mathbf{f}$ conditional on $\mathcal{F}_n$. As introduced in Kallenberg (2002, p. 109) we denote this by $X_{n+1} \perp\!\!\!\perp_{\mathcal{F}_n} \mathbf{f}$. In our setting conditional independence is equivalent to[3] $X_{n+1} = h(\xi, U)$ for a measurable function $h$, a random (previsible) element $\xi$ that generates $\mathcal{F}_n$ and a standard uniform random variable $U$ independent from $(\mathbf{f}, \mathcal{F}_n)$ (Kallenberg, 2002, Prop. 6.13).

**Noisy evaluations**   In many optimization applications only noisy evaluations of the random objective function $\mathbf{f}$ at $x$ may be obtained. We associate the noise $\varsigma_n$ to the $n$-th evaluation $x_n$, such that the function $\mathbf{f}_n = \mathbf{f} + \varsigma_n$ returns the $n$-th observation $Y_n = \mathbf{f}_n(x_n)$. While the noise may simply be independent, identically distributed constants, observe that this framework allows for much more general location-dependent noise. The only requirement is that the random noise functions $\varsigma_n$ are continuous, such that the $\mathbf{f}_n$ are continuous functions.

**Definition 2.7** (Conditionally independent evolution). The *general conditional independence setting* is given by

  • an underlying probability space $(\Omega, \mathcal{A}, \mathbb{P})$,
  • a sub-$\sigma$-algebra $\mathcal{F}$ (the 'initial information') with $W$ a random element such that $\mathcal{F} = \sigma(W)$,[4]

---

[3]The equivalence only requires $X_{n+1}$ to be a random variable in a standard Borel space.
[4]There always exists such a random element $W$ since the identity map from the measurable space $(\Omega, \mathcal{A})$ into $(\Omega, \mathcal{F})$ is measurable and clearly generates $\mathcal{F}$.

- A sequence $(\mathbf{f}_n)_{n\in\mathbb{N}_0}$ of continuous random functions with $\mathbf{f}_n$ in $C(\mathbb{X}_n, \mathbb{Y}_n)$, where the domains $(\mathbb{X}_n)_{n\in\mathbb{N}_0}$ are locally compact, separable metrizable spaces and the codomains $(\mathbb{Y}_n)_{n\in\mathbb{N}_0}$ Polish spaces.
- a random variable $Z$ in a standard Borel space $(E, \mathcal{B}(E))$ (representing an additional quantity of interest).

A sequence $X = (X_n)_{n\in\mathbb{N}_0}$ of random evaluation locations with $X_n \in \mathbb{X}_n$ is called a *conditionally independent evolution*, if $X_{n+1} \perp\!\!\!\perp_{\mathcal{F}_n} (Z, (\mathbf{f}_n)_{n\in\mathbb{N}_0})$ for the filtration

$$\mathcal{F}_n := \sigma(\mathcal{F}, \mathbf{f}_0(X_0), \dots, \mathbf{f}_n(X_n), X_{[0:n]}) \qquad \text{for } n \geq 0, \qquad \mathcal{F}_{-1} := \mathcal{F}, \tag{4}$$

where $x_I = (x_i)_{i\in I}$ and we introduce compact notation for *discrete intervals*:

$$[i\!:\!j] := [i, j] \cap \mathbb{Z}, \qquad [i\!:\!j) := [i, j) \cap \mathbb{Z}, \qquad \text{etc.} \qquad\qquad \text{(discrete intervals)}$$

In the following we generalize the definition of PIC to this conditional independence setting before we state our main result.

**Definition 2.8** (Plug-in consistent – PIC). Let $\mathbb{X} = \prod_{k=0}^n \mathbb{X}_k$ and let the collection of kernels $(\kappa_{x_{[0:n]}})_{x_{[0:n]}\in\mathbb{X}}$ with

$$\mathbb{P}\big((Z, \mathbf{f}_n(x_n)) \in A \mid \mathcal{F}, (\mathbf{f}_k(x_k))_{k\in[0:n)}\big) \overset{\text{a.s.}}{=} \kappa\big(W, (\mathbf{f}_k(x_k))_{k\in[0:n)}, x_{[0:n]}; A\big)$$

form a joint kernel $\kappa$ (Definition 2.1). Then this joint kernel $\kappa$ is called *plug-in consistent* (PIC), if **for all** conditionally independent evolutions $(X_k)_{k\in\mathbb{X}}$ and their filtration $\mathcal{F}_n$ as defined in (4), we have

$$\mathbb{P}\big((Z, \mathbf{f}_n(X_n)) \in A \mid \mathcal{F}_{n-1}, X_n\big) \overset{\text{a.s.}}{=} \kappa\big(W, (\mathbf{f}_k(X_k))_{k\in[0:n)}, X_{[0:n]}; A\big).$$

*Remark* 2.9 (This is a generalization). Note that PIC as defined above is a generalization of Definition 2.3 with $n = 0$ and $X = X_0$.

In our main result we allow for random evaluations in the conditional. This is significantly harder to prove than random evaluation locations in the dependent part of the conditional distribution. The main ingredient of the proof is a general result that allows moving plug-in consistency (PIC) from the dependent to the conditional part (Proposition 4.2). We iteratively combine this result with Proposition 2.4 that only allows random variables in the dependent part.

**Theorem 2.10** (Conditionally independent sampling). *Assume the general conditional independence setting (Definition 2.7).*

(i) **Without the dependent variable, every joint conditional distribution is PIC.** *That is, let $\kappa$ be a joint conditional distribution for $Z$ given $(\mathcal{F}, \mathbf{f}_0(x_0), \dots, \mathbf{f}_n(x_n))$, that is for all $x_{[0:n]} \in \prod_{k=0}^n \mathbb{X}_k$ and $A \in \mathcal{B}(E)$*

$$\mathbb{P}\big(Z \in A \mid \mathcal{F}, \mathbf{f}_0(x_0), \dots, \mathbf{f}_n(x_n)\big) \overset{a.s.}{=} \kappa\big(W, \mathbf{f}_0(x_0), \dots, \mathbf{f}_n(x_n), x_{[0:n]}; A\big).$$

*Then for all conditionally independent evolutions $(X_k)_{k\in\mathbb{N}_0}$ and $A \in \mathcal{B}(E)$*

$$\mathbb{P}\big(Z \in A \mid \mathcal{F}_n\big) \overset{a.s.}{=} \kappa\big(W, \mathbf{f}_0(X_0), \dots, \mathbf{f}_n(X_n), X_{[0:n]}; A\big).$$

(ii) **With the dependent variable, continuity is sufficient for joint conditional distributions to be PIC.** *That is, let the kernel $\kappa$ be a joint conditional distribution for $(Z, \mathbf{f}_n(x_n))$ given $(\mathcal{F}, (\mathbf{f}_k(x_k))_{k\in[0:n)})$ such that*

$$x_n \mapsto \kappa(y_{[0:n)}, x_{[0:n]}; \cdot)$$

*is continuous in the weak topology for all $x_{[0:n)} \in \mathbb{X}^n$ and all $y_{[0:n)} \in \mathbb{Y}^n$. Then for all conditionally independent evolutions $(X_k)_{k\in\mathbb{N}_0}$ and measurable sets $B \in \mathcal{B}(E) \otimes \mathcal{B}(\mathbb{Y})$*

$$\mathbb{P}\big((Z, \mathbf{f}_n(X_n)) \in B \mid \mathcal{F}_{n-1}, X_n\big) \overset{a.s.}{=} \kappa\big(W, (\mathbf{f}_k(X_k))_{k\in[0:n)}, X_{[0:n]}; B\big).$$

*Remark* 2.11 (Comparison with dependent evaluation). We highlight that continuity of the kernel is only required for the case where function evaluations are dependent variables. However, while Proposition 2.4 shows the existence of a joint kernel that admits random evaluation, the existence of such a joint conditional distribution is not proven here. In the Gaussian case this is not a problem, since the joint conditional distribution is known explicitly (see Section A).

**Corollary 2.12** (Gaussian case). *In the general conditional independence setting (Definition 2.7) further assume that the sequence $(\mathbf{f}_n)_{n \in \mathbb{N}_0}$ consists of joint Gaussian random functions. Then the canonical Gaussian conditional distribution (Definition A.3) for $\mathbf{f}(x, n) = \mathbf{f}_n(x)$ is PIC.*

*Proof.* The canonical Gaussian conditional distribution is a joint conditional distribution (Remarks A.4) that also satisfies the continuity requirement in Theorem 2.10 (ii) (Remark A.5). □

### 2.3 Counterexample

While it is unlikely that anyone would question the measurability requirement in Corollary 2.5, the continuity requirement may seem strange. Especially, as we do not appear to require it for the evaluation of functions in the conditional (see Theorem 2.10). The following counterexample illustrates that such a continuity requirement is indeed necessary for the PIC property.

**Counterexample 2.13** (Joint kernel that is not PIC). In this example we show that $X$ measurable with respect to $\mathcal{F}$ is not always sufficient to treat $X$ as deterministic in $\mathbb{E}[\mathbf{f}(X) \mid \mathcal{F}]$.

For this purpose consider a standard normal random variable $Y \sim \mathcal{N}(0, 1)$ independent of a standard uniform random variable $U \sim \mathcal{U}(0, 1)$. We define an almost surely constant Gaussian random function $\mathbf{f}(x) := Y$ and define $\mathcal{F} := \sigma(U)$. Since the conditional expectation is only well-defined up to null sets we have

$$\mathbb{E}[\mathbf{f}(x) \mid U] \stackrel{\text{a.s.}}{=} \mathbb{E}[Y] \mathbf{1}_{U \neq x} =: g(x).$$

However we have

$$g(U) = 0 \neq \mathbb{E}[Y] \stackrel{\text{a.s.}}{=} \mathbb{E}[\mathbf{f}(U) \mid U].$$

While the formula $g$ is therefore a valid formula for $\mathbb{E}[\mathbf{f}(x) \mid U]$ for all deterministic $x$, it is not valid for random $X = U$. Even though $X$ is measurable with respect to $\mathcal{F} = \sigma(U)$. Similarly one could construct $g$ to not be a measurable function. In this case $g(U)$ would not even be a valid random variable. This further justifies (i) in Corollary 2.5.

*Remark* 2.14 (Connection to regular conditional probability). In the Counterexample above we created different adversarial null sets for each $x$, which joined together in the case of the random $X = U$ to break the formula. Similar counterexamples motivate the construction of the *regular conditional probability kernel*[5] (e.g. Klenke, 2014, Def. 8.28). Just like regular conditional probabilities result in a narrower definition of a "sensible" conditional probability, PIC joint conditional distributions further narrow the set of conditional distributions down.

## 3 Applications

In the case of Gaussian random functions $\mathbf{f}$ for example, $(\mathbf{f}(x_0), \ldots, \mathbf{f}(x_n))$ is a multivariate Gaussian vector with well known posterior distribution $\mathbf{f}(x_n)$ given $(\mathbf{f}(x_0), \ldots, \mathbf{f}(x_{n-1}))$ when the evaluation locations are deterministic. But $\mathbf{f}(X)$ is not necessarily Gaussian if $X$ is random[6] and the calculation of conditional distributions hence becomes much more difficult. Treating previsible inputs as deterministic ensures the calculation is feasible but it requires a theoretical foundation.

---

[5]The conditional probability $\mathbb{P}(X \in A \mid \mathcal{F}) = \mathbb{E}[\mathbf{1}_A \mid \mathcal{F}]$ for a random variable $X$, $\sigma$-algebra $\mathcal{F}$ and measurable sets $A$ is only well defined up to a null set. It is therefore ex ante impossible to ensure that $A \mapsto \mathbb{P}(X \in A \mid \mathcal{F})$ is a well-defined measure. This is because the null-set may depend on $A$ and, if adversarially selected, their union may no longer be a null set.

[6]consider $X = \arg\min_{x \in K} \mathbf{f}(x)$ for some compact set $K \subseteq \mathbb{X}$.

### 3.1 Maximal probability of improvement

To illustrate the necessity of our results, we consider a simple optimization procedure from Bayesian optimization known as "maximal probability of improvement" (PI) (e.g. Garnett, 2023, Sec. 7.5).

Let $(\mathbf{f}(x))_{x \in \mathbb{X}}$ be a Gaussian random function with values in $\mathbb{R}$. The goal is to find a maximum of $\mathbf{f}$. To this end, new evaluation locations are chosen according to the rule

$$X_n := \underset{x_n \in \mathbb{X}}{\arg\max} \, \mathbb{P}\Big(\mathbf{f}(x_n) > \underbrace{\max_{i=0,\dots,n-1} \mathbf{f}(X_i) + \epsilon}_{=:\eta} \mid \mathcal{F}_{n-1}\Big), \tag{PI}$$

where $\epsilon > 0$ is a minimum improvement and $\mathcal{F}_n = \sigma(\mathbf{f}(X_0), \dots, \mathbf{f}(X_n))$. And we assume a deterministic starting location $X_0 = x_0$. We will address the measure-theoretic subtleties of maximization in Subsection 3.1.2, let us first consider whether or not we can even obtain an explicit expression for the probability of improvement.

#### 3.1.1 Calculating the probability of improvement

**Deterministic case**  Computing PI explicitly would be straightforward if the $X_0, \dots, X_{n-1}$ were deterministic $x_0, \dots, x_{n-1}$. In this case $(\mathbf{f}(x_0), \dots, \mathbf{f}(x_n))$ is a multivariate Gaussian random vector. Consequently, the conditional distribution of $\mathbf{f}(x_n)$ given $(\mathbf{f}(x_0), \dots, \mathbf{f}(x_{n-1}))$ is again Gaussian (see Lemma A.2). Let $\Phi$ denote the cumulative distribution function of the standard normal distribution $\mathcal{N}(0,1)$. Since $\eta$ is $\mathcal{F}_{n-1}$ measurable we have

$$\mathbb{P}\big(\mathbf{f}(x_n) > \eta \mid \mathcal{F}_{n-1}\big) = 1 - \Phi\big(\tfrac{\eta - \mu_n}{\sigma_n}\big),$$

where the posterior mean and variance,

$$\mu_n = \mu_n(x_0, \dots, x_n, \mathbf{f}(x_0), \dots, \mathbf{f}(x_{n-1})) \quad \text{and} \quad \sigma_n^2 = \sigma_n^2(x_0, \dots, x_n),$$

are given explicitly in Lemma A.2. Since $\Phi$ is monotonically increasing this results in the optimization problem

$$X_n = \underset{x_n}{\arg\max} \, \mathbb{P}\big(\mathbf{f}(x_n) > \eta \mid \mathcal{F}_{n-1}\big) = \underset{x_n}{\arg\max} \, \frac{\mu_n - \eta}{\sigma_n}, \tag{5}$$

which can be numerically maximized using the explicit formulas in Lemma A.2.

**Heuristic in the random case**  With the heuristic motivation that the $X_k$ are 'deterministic' conditioned on $\mathcal{F}_{k-1}$ ("previsible"), the same procedure is used when the $X_0, \dots, X_{n-1}$ are *not* deterministic but selected by this process. The correctness of this procedure is often treated as self-evident (e.g. Srinivas et al., 2012, Lemma 5.1, p. 3258). A proof of this conjecture is non-trivial but a direct result of Corollary 2.12.

**Example 3.1** (Calculating the probability of improvement)**.**  Let $\mathbf{f}$ be a continuous Gaussian random function, $X_k$ measurable with respect to $\mathcal{F}_{k-1} = \sigma(\mathbf{f}(X_0), \dots, \mathbf{f}(X_{k-1}))$ and $X_0 = x_0$ deterministic. Then, if the probability of improvement

$$\mathbb{P}\Big(\mathbf{f}(x_n) > \max_{k \in [0:n)} \mathbf{f}(X_k) + \epsilon \mid \mathcal{F}_{n-1}\Big)$$

is calculated using the canonical Gaussian conditional distribution (Definition A.3) the random evaluation locations $X_k$ may be treated like deterministic locations $x_k$.

*Proof.*  Define the continuous function

$$h(y_{[0:n]}) := y_n - \max_{k \in [0:n)} y_k.$$

Using $\mathbf{f}_n(x_{[0:n]}) := (\mathbf{f}(x_0), \dots, \mathbf{f}(x_n))$ the desired probability is then clearly

$$\mathbb{P}\Big(\mathbf{f}_n(X_{[0:n)}, x_n) \in h^{-1}((\epsilon, \infty)) \mid \mathbf{f}(X_0), \dots, \mathbf{f}(X_{n-1})\Big).$$

Since the $(\mathbf{f}_k)_{k \leq n}$ with $\mathbf{f}_k := \mathbf{f}$ for $k < n$ are joint Gaussian, the canonical Gaussian conditional distribution (Definition A.3) for deterministic locations forms a joint kernel $\kappa$ (Remark A.4) with

$$\mathbb{P}\Big(\mathbf{f}_n(\tilde{x}_n) \in A \mid \mathbf{f}(x_0), \ldots, \mathbf{f}(x_{n-1})\Big) = \kappa(\mathbf{f}(x_0), \ldots, \mathbf{f}(x_{n-1}), x_{[0:n)}, \tilde{x}_n; A),$$

We may then apply Corollary 2.12 to obtain that we have for $\tilde{X}_n = (X_{[0:n)}, x_n)$

$$\mathbb{P}\Big(\mathbf{f}(x_n) > \max_{k \in [0:n)} \mathbf{f}(X_k) + \epsilon \mid \mathcal{F}_{n-1}\Big) = \kappa(\mathbf{f}(X_0), \ldots, \mathbf{f}(X_{n-1}), X_{[0:n)}, \tilde{X}_n; A).$$

Or in other words, we may treat the inputs $X_0, \ldots, X_{n-1}$ like deterministic parameters in the calculation. □

### 3.1.2 Subtleties of maximization

Assuming there exists a continuous version of $x \mapsto \mathbb{P}(\mathbf{f}(x) > \eta \mid \mathcal{F}_{n-1})$ and $\mathbb{X}$ is compact, the maximizer $X_n$ in PI exists. And it is perhaps reasonable to expect

$$\mathbb{P}\big(\mathbf{f}(X_n) > \eta \mid \mathcal{F}_{n-1}\big) \stackrel{\text{a.s.}}{=} \max_{x_n \in \mathbb{X}} \mathbb{P}\big(\mathbf{f}(x_n) > \eta \mid \mathcal{F}_{n-1}\big), \tag{6}$$

since $X_n$ is defined as the maximizer. However, the supremum of a random function is well-known to be a subtle measure-theoretic object that is only well defined if the random function is separable. The conditional probabilities are such random functions in $x_n$ (assuming they are formed using a joint kernel for measurability). The following illustrative example explains why we cannot expect (6) in general.

**Example 3.2** (Maximization of conditional expectations). Let $U \sim \mathcal{U}([0, 1])$ and $Y \sim \mathcal{N}(0, 1)$ be independent random variables and define $\mathbf{f}(x) = Y$ to be the constant Gaussian random function over $x \in [0, 1]$. Since the conditional expectation is only well defined up to null sets we clearly have for $\gamma \gg \mathbb{E}[Y]$

$$\mathbb{E}[\mathbf{f}(x) \mid U] \stackrel{\text{a.s.}}{=} \mathbb{E}[Y] + \mathbf{1}_{x=U}\gamma =: g(x),$$

as $\{x = U\}$ is a null set. Therefore we have

$$\max_{x \in [0,1]} g(x) = \gamma, \qquad \arg\max_{x \in [0,1]} g(x) = U.$$

However this does not translate back

$$\mathbb{E}[\mathbf{f}(U) \mid U] \stackrel{\text{a.s.}}{=} \mathbb{E}[Y] \neq g(U) = \gamma.$$

Since $g(x)$ is a valid version of $\mathbb{E}[\mathbf{f}(x) \mid U]$ the term

$$\sup_{x \in [0,1]} \mathbb{E}[\mathbf{f}(x) \mid U]$$

is not well defined if we admit such $g$. However, if we require $x \mapsto \mathbb{E}[\mathbf{f}(x) \mid U]$ to be separable (e.g. continuous), then the supremum is well defined.

We now may want to prove (6) for separable versions of the conditional distribution with Proposition 2.4. For this we require the following about the PIC joint kernel for all $x_n \in \mathbb{X}$ and all $\eta \in \mathbb{R}$

$$\mathbb{P}\big(\mathbf{f}(x_n) = \eta \mid \mathcal{F}_{n-1}\big) \stackrel{\text{a.s.}}{=} \kappa(\cdot, x_n; \{\eta\}) = 0. \tag{7}$$

This is generally the case for Gaussian random functions, since the value

$$\eta = \max_{i=0,\ldots,n-1} \mathbf{f}(X_i) + \epsilon$$

is not achieved at any of the locations $X_i$ for $\epsilon > 0$ and all other points typically have positive remaining variance.

**Example 3.3** (Maximizer plugged into PI). Assuming (7) we have

$$\mathbb{P}\big(\mathbf{f}(X_n) > \eta \mid \mathcal{F}_{n-1}\big) \stackrel{\text{a.s.}}{=} \max_{\substack{X \ \mathcal{F}_{n-1}\text{-meas.} \\ \text{r.v. in } \mathbb{X}}} \mathbb{P}\big(\mathbf{f}(X) > \eta \mid \mathcal{F}_{n-1}\big).$$

$$\stackrel{\text{a.s.}}{=} \max_{x_n \in \mathbb{X}} \mathbb{P}\big(\mathbf{f}(x_n) > \eta \mid \mathcal{F}_{n-1}\big).$$

*Proof.* Using $h(y_{[0:n]}) = y_n - \max_{k \in [0:n]} y_k$ and $\mathbf{f}_n(x_{[0:n]}) := (\mathbf{f}(x_0), \dots, \mathbf{f}(x_n))$ again, we have by Proposition 2.4 that there exists a kernel with

$$\mathbb{P}(\mathbf{f}_n(X_{[0:n)}, x_n) \in A \mid \mathcal{F}_{n-1})(\omega) = \kappa(\omega, X_{[0:n)}, x_n; A),$$

where $\tilde{x} \mapsto \kappa(\omega, \tilde{x}; \cdot)$ is continuous in the weak topology of measures. Since $(\epsilon, \infty)$ is a continuity set, i.e. $\mathbb{P}(\mathbf{f}_n(X_{[0:n)}, x_n) = \epsilon \mid \mathcal{F}_{n-1}) = 0$ by (7), we have by the Portmanteau theorem (e.g. Klenke, 2014, Thm. 13.16) that the following function is continuous

$$x_n \mapsto \kappa(\omega, X_{[0:n)}, x_n; (\epsilon, \infty)).$$

This function is thereby a separable version of PI and thus

$$\max_{x_n \in \mathbb{X}} \mathbb{P}\big(\mathbf{f}(x_n) > \eta \mid \mathcal{F}_{n-1}\big) \stackrel{\text{a.s.}}{=} \max_{x_n \in \mathbb{X}} \kappa(\cdot, X_{[0:n)}, x_n; [\epsilon, \infty))$$

$$\geq \kappa(\cdot, X_{[0:n)}, X_n; [\epsilon, \infty))$$

$$\stackrel{\text{a.s.}}{=} \mathbb{P}\big(\mathbf{f}(X_n) > \eta \mid \mathcal{F}_{n-1}\big)$$

for any $X_n$ that is $\mathcal{F}_{n-1}$-measurable and takes values in $\mathbb{X}$. Taking the maximum over such $X_n$ results in the second equality since deterministic $x_n$ are special cases. Since

$$X_n = \arg\max_{x_n} \mathbb{P}\big(\mathbf{f}(x_n) > \eta \mid \mathcal{F}_{n-1}\big) \stackrel{\text{a.s.}}{=} \arg\max_{x_n} \kappa(\cdot, X_{[0:n)}, x_n; [\epsilon, \infty)) =: \tilde{X}_n,$$

we clearly also have the first equality

$$\mathbb{P}\big(\mathbf{f}(X_n) > \eta \mid \mathcal{F}_{n-1}\big) \stackrel{\text{a.s.}}{=} \kappa(\cdot, X_{[0:n)}, X_n; [\epsilon, \infty))$$

$$\stackrel{\text{a.s.}}{=} \kappa(\cdot, X_{[0:n)}, \tilde{X}_n; [\epsilon, \infty))$$

$$= \max_{x_n \in \mathbb{X}} \kappa(\cdot, X_{[0:n)}, x_n; [\epsilon, \infty))$$

$$\stackrel{\text{a.s.}}{=} \max_{x_n \in \mathbb{X}} \mathbb{P}\big(\mathbf{f}(x_n) > \eta \mid \mathcal{F}_{n-1}\big). \qquad \square$$

*Remark* 3.4. It may be possible to get rid of the assumption (7) if one can combine the continuity of the kernel in the weak sense with the fact that the function $x \mapsto \mathbb{P}(\mathbf{f}(x) > \eta \mid \mathcal{F}_{n-1})$ for the fixed set $[\epsilon, \infty)$ must be separable.

## 3.2 Maximal expected improvement

Another popular method in Bayesian optimization is the maximization of the *expected improvement* (EI) (e.g. Garnett, 2023, Sec. 7.3). Here the next evaluation location is chosen as

$$X_n = \arg\max_{x_n \in \mathbb{X}} \mathbb{E}\Big[\big(\mathbf{f}(x_n) - \mathbf{f}_n^*\big)_+ \mid \mathcal{F}_{n-1}\Big].$$

for $\mathbf{f}_n^* = \max_{k \in [0:n)} \mathbf{f}(X_k)$ with $x_+ = \max\{0, x\}$, assuming $\sup_{x \in \mathbb{X}} \mathbb{E}|\mathbf{f}(x)| < \infty$.

**Example 3.5** (Expected improvement with previsible inputs). Let $\mathbf{f}$ be a continuous Gaussian random function and $X_k$ measurable with respect to the filtration $\mathcal{F}_{k-1} = \sigma(\mathbf{f}(X_0), \dots, \mathbf{f}(X_{k-1}))$ and $X_0 = x_0$ deterministic. Then, the expected improvement

$$\mathbb{E}\Big[\big(\mathbf{f}(x_n) - \mathbf{f}_n^*\big)_+ \mid \mathcal{F}_{n-1}\Big]$$

may be calculated using the canonical conditional distribution (Definition A.3) by treating the random evaluation locations $X_k$ like deterministic locations $x_k$.

*Proof.* Using the continuous function

$$h(y_{[0:n]}) = (y_n - \max_{k \in [0:n]} y_k)_+$$

and $\mathbf{f}_n(x) = (\mathbf{f}(x_0), \dots, \mathbf{f}(x_n))$ the expected improvement can be written as

$$\mathbb{E}\Big[h \circ \mathbf{f}_n(X_{[0:n)}, x_n) \mid \mathcal{F}_{n-1}\Big] = \int h(y)\, \mathbb{P}(\mathbf{f}_n(X_{[0:n)}, x_n) \in dy \mid \mathcal{F}_{n-1})$$

Application of Corollary 2.12 to $\mathbb{P}(\mathbf{f}_n(X_{[0:n)}, x_n) \in dy \mid \mathcal{F}_{n-1})$ analogous to the proof of Example 3.1 yields the claim. □

Without the assumption $\mathbb{E} \sup_{x \in \mathbb{X}} |\mathbf{f}(x)| < \infty$ it is already difficult to justify the existence of a continuous conditional expectation $x \mapsto \mathbb{E}[\mathbf{f}(x) \mid \mathcal{F}]$. The following assumption in the example on maximization is therefore very natural.

**Example 3.6** (Maximizers plugged into expected improvement)**.** Assuming $\mathbb{E}[\sup_{x \in \mathbb{X}} |\mathbf{f}(x)|] < \infty$ we have

$$\mathbb{E}\Big[\big(\mathbf{f}(X_n) - \mathbf{f}_n^*\big)_+ \mid \mathcal{F}_{n-1}\Big] = \max_{\substack{X\ \mathcal{F}_{n-1}\text{-meas.} \\ \text{r.v. in } \mathbb{X}}} \mathbb{E}\Big[\big(\mathbf{f}(X) - \mathbf{f}_n^*\big)_+ \mid \mathcal{F}_{n-1}\Big]$$

$$= \max_{x_n \in \mathbb{X}} \mathbb{E}\Big[\big(\mathbf{f}(x_n) - \mathbf{f}_n^*\big)_+ \mid \mathcal{F}_{n-1}\Big].$$

*Proof.* Using $h(y_{[0:n]}) = (y_n - \max_{k \in [0:n)} y_k)_+$ and $\mathbf{f}_n$ again, we have

$$\mathbb{E}[\sup_{x \in \mathbb{X}} |h \circ \mathbf{f}_n(x_{[0:n]})| \mid \mathcal{F}_{n-1}] \le (n+1)\mathbb{E}\Big[\sup_{x \in \mathbb{X}} |\mathbf{f}(x)| \mid \mathcal{F}_n\Big] < \infty$$

almost surely. For the kernel $\kappa$ from Proposition 2.4 with

$$\mathbb{P}(\mathbf{f}_n(x_{[0:n)}, x_n) \in A \mid \mathcal{F}_{n-1}) = \kappa(\cdot, x_{[0:n)}, x_n; A)$$

we thereby know that

$$H(x_n) := \int h(y)\, \kappa(\cdot, X_{[0:n)}, x_n; dy) \overset{\text{a.s.}}{=} \mathbb{E}\Big[\big(\mathbf{f}(x_n) - \mathbf{f}_n^*\big)_+ \mid \mathcal{F}_{n-1}\Big]$$

is an almost surely continuous version of the expected improvement into which we may plug random $X_n$ that are measurable with respect to $\mathcal{F}_{n-1}$. The rest of the proof is then analogous to the proof of Example 3.3. □

## 4 Proofs

### 4.1 Simplified case: Dependent evaluation

*Proof of Proposition 2.4.* Observe that $E \times C(\mathbb{X}, \mathbb{Y})$ is a standard borel space since $C(\mathbb{X}, \mathbb{Y})$ is Polish (Theorem B.1). There therefore exists a regular conditional probability distribution $\kappa_{Z,\mathbf{f}|\mathcal{F}}$ (e.g. Kallenberg, 2002, Thm. 6.3). Using this probability kernel, we define the kernel

$$\kappa(\omega, x; B) := \int \mathbf{1}_B(z, e(f, x)) \kappa_{Z,\mathbf{f}|\mathcal{F}}(\omega; dz \otimes df)$$

which is a measure in $B \in \mathcal{B}(E) \otimes \mathcal{B}(\mathbb{Y})$ by linearity of the integral and monotone convergence. We therefore only need to prove it is measurable in $(\omega, x) \in \Omega \times \mathbb{X}$ to prove it is a probability kernel. This follows from measurability of the evaluation function $e$ (Theorem B.1) and the application of Lemma 14.20 by Klenke (2014) to the probability kernel $\tilde{\kappa}(\omega, x; A) := \kappa_{Z,\mathbf{f}|\mathcal{F}}(\omega; A)$ in the equation above. By 'disintegration' (e.g.

Kallenberg, 2002, Thm 6.4) this probability kernel is moreover a regular conditional version of $\mathbb{P}((Z, \mathbf{f}(X)) \in B \mid \mathcal{F})$ for all $\mathcal{F}$-measurable $X$, i.e. for all $B \in \mathcal{B}(E) \otimes \mathcal{B}(\mathbb{Y})$ and for $\mathbb{P}$-almost all $\omega$

$$\mathbb{P}\big((Z, \mathbf{f}(X)) \in B \mid \mathcal{F}\big)(\omega) \overset{\text{disint.}}{=} \int \mathbf{1}_B(z, e(f, X(\omega)))\kappa_{Z,\mathbf{f}\mid\mathcal{F}}(\omega; dz \otimes df)$$

$$\overset{\text{def.}}{=} \kappa(\omega, X(\omega); B).$$

The kernel thereby satisfies (2).

For continuity observe that we have $\lim_{x \to y}(z, f(x)) = (z, f(y))$ for any $f \in C(\mathbb{X}, \mathbb{Y})$. For open $U$ this implies

$$\liminf_{x \to y} \mathbf{1}_U(z, f(x)) \geq \mathbf{1}_U(z, f(y)),$$

because if $(z, f(y)) \in U$, then eventually $(z, f(x))$ in $U$ due to openness of $U$. An application of Fatou's lemma (e.g. Klenke, 2014, Thm. 4.21) yields for all open $U$

$$\liminf_{x \to y} \kappa(\omega, x; U) \geq \int \liminf_{x \to y} \mathbf{1}_U(z, f(x))\kappa_{Z,\mathbf{f}\mid\mathcal{F}}(\omega; dz \otimes df) \geq \kappa(\omega, y; U).$$

And we can conclude weak convergence by the Portmanteau theorem (Klenke, 2014, Thm. 13.16) since $E \times \mathbb{Y}$ is metrizable.

Let $\tilde{\kappa}$ be another joint probability kernel that is continuous almost surely. We will assume it is continuous for all $\omega$ without loss of generality in the following and assume that the null set of discontinuity is tacitly joined with the null set we construct. Since $E \times \mathbb{Y}$ is second countable, there is a countable base $\{U_n\}_{n \in \mathbb{N}}$ of its topology, which generates the Borel $\sigma$-algebra $\mathcal{B}(E) \otimes \mathcal{B}(\mathbb{Y})$. And since $\mathbb{X}$ is separable, it has a countable dense subset $Q$. There must therefore exist a null set $N$ such that

$$\kappa(\omega, q; U_n) = \tilde{\kappa}(\omega, q; U_n), \qquad \forall \omega \in N^\complement, \ n \in \mathbb{N}, \ q \in Q,$$

because both kernels are regular conditional version of $\mathbb{P}(Z, \mathbf{f}(q) \in U_n; \mathcal{G})$ and the union over $\mathbb{N} \times Q$ is a countable union. Since $\{U_n\}_{n \in \mathbb{N}}$ generates the $\sigma$-algebra, we deduce for all $\omega \in N^\complement$ and all $q \in Q$ that $\kappa(\omega, q; \cdot) = \tilde{\kappa}(\omega, q; \cdot)$. As $Q$ is dense in $\mathbb{X}$ we have by continuity of the joint kernels for all $\omega \in N^\complement$ and all $x \in \mathbb{X}$

$$\kappa(\omega, x; \cdot) = \tilde{\kappa}(\omega, x; \cdot).$$

Let $g \colon E \times \mathbb{Y} \to \mathbb{R}$ be a continuous function with $\mathbb{E}[\sup_x |g(Z, \mathbf{f}(x))| \mid \mathcal{F}] < \infty$ almost surely. Then for $\mathbf{g}(x) = \int g(z, y)\kappa(\cdot, x; dz \otimes dy)$ we have

$$|\mathbf{g}(x) - \mathbf{g}(x_0)|(\omega) = \left| \int g(z, y)\kappa(\omega, x; dz \otimes dy) - \int g(z, y)\kappa(\omega, x_0; dz \otimes dy) \right|$$

$$= \left| \int g(z, e(f, x)) - g(z, e(f, x_0))\kappa_{Z,\mathbf{f}\mid\mathcal{F}}(\omega; dz \otimes df) \right|$$

$$\leq \int \big| g(z, f(x)) - g(z, f(x_0)) \big| \kappa_{Z,\mathbf{f}\mid\mathcal{F}}(\omega; dz \otimes df)$$

using the definition of $\kappa$ via $\kappa_{Z,\mathbf{f}\mid\mathcal{F}}$ for the second equation. Up to a null set $N$ we further have for all $\omega \in N^\complement$

$$\int \sup_x |g(z, f(x))|\kappa_{Z,\mathbf{f}\mid\mathcal{F}}(\omega; dz \otimes df) = \mathbb{E}\Big[\sup_x |g(Z, \mathbf{f}(x))| \ \Big| \ \mathcal{F}\Big](\omega) < \infty.$$

Since $|g(z, f(x)) - g(z, f(x_0))| \leq 2\sup_x |g(z, \mathbf{f}(x))|$ we thus have by dominated convergence and continuity of $g$ and $\mathbf{f}$ that $\mathbf{g}(x) \to \mathbf{g}(x_0)$ for $x \to x_0$. $\qquad\square$

*Remark* 4.1 (Possible generalization). Note that for the existence of a joint conditional distribution that satisfies (2) we only require a regular conditional distribution for $(Z, \mathbf{f})$ given $\mathcal{F}$ to exist and measurability of the evaluation map $e$. This part of the result can therefore be made to hold with greater generality.

## 4.2 General case: Conditionally independent evaluation locations

To prove our main result (Theorem 2.10) we will move the PIC property that we can obtain by Proposition 2.4 for function values in the dependent part to the conditional part. The general proposition we will prove in the following is the key tool for this purpose. We change the notation of the random objects to highlight that we no longer assume continuity or make assumptions about the domain and range. This proposition essentially states:

> Let $\xi_1, \xi_3$ be random elements and $\xi_2 = (\xi_2^y)_y$ be a collection of random elements indexed by $y$. If there exists a PIC joint conditional distribution $\kappa_{3,2|1}$ for $(\xi_3, \xi_2^y)$ given $\xi_1$, then **any** joint conditional distribution $\kappa_{3|2,1}$ for $\xi_3$ given $(\xi_2^y, \xi_1)$ is PIC.

The "joint" in "joint conditional distribution" refers to the index $y$ of $\xi_2^y$. The formal statement of this claim follows:

**Proposition 4.2** (Consistency shuffle). *Let $(\Omega, \mathcal{A}, \mathbb{P})$ be a probability space and let $\xi_1, (\xi_2^y)_{y \in D}, \xi_3$ be random elements in the measurable spaces $(E_i, \mathcal{E}_i)$, $i \in \{1, 2, 3\}$, where $\xi_2$ is indexed by the measurable domain $(D, \mathcal{D})$. Assume $\xi_2^{g(\xi_1)}$ is measurable as a random element in $\omega \in \Omega$ for all measurable functions $g \colon E_1 \to D$, and assume there exists a PIC joint conditional distribution $\kappa_{3,2|1}$ for $\xi_3, \xi_2^y$ given $\xi_1$. That is for all $A \in \mathcal{E}_3 \otimes \mathcal{E}_2$ and all measurable functions $g \colon E_1 \to D$*

$$\mathbb{P}\big(\xi_3, \xi_2^{g(\xi_1)} \in A \mid \xi_1\big) \overset{a.s.}{=} \kappa_{3,2|1}(\xi_1, g(\xi_1); A). \tag{8}$$

*If there exists a joint conditional probability kernel $\kappa_{3|1,2}$ for $\xi_3$ given $\xi_1, \xi_2^y$ such that for all $y \in D$ and $A_3 \in \mathcal{E}_3$*

$$\mathbb{P}(\xi_3 \in A_3 \mid \xi_1, \xi_2^y) \overset{a.s.}{=} \kappa_{3|2,1}(\xi_1, \xi_2^y, y; A_3), \tag{9}$$

*then $\kappa_{3|2,1}$ is PIC, that is we have for all $A_3 \in \mathcal{E}_3$ and measurable $g \colon E_1 \to D$*

$$\mathbb{P}(\xi_3 \in A_3 \mid \xi_1, \xi_2^{g(\xi_1)}) \overset{a.s.}{=} \kappa_{3|2,1}\big(\xi_1, \xi_2^{g(\xi_1)}, g(\xi_1); A_3\big).$$

*Remark* 4.3 (Possible generalization). Note that we keep $g$ fixed throughout the proof. So if $\kappa_{3,2|1}$ is PIC only for a specific $g$, then we also obtain consistency of $\kappa_{3|2,1}$ only for this specific function $g$. For consistency of $\kappa_{3|2,1}$ it is therefore sufficient to find a $\kappa_{3,2|1}^g$ that is only PIC w.r.t. $g$ for each $g$.

*Proof.* Let $g \colon E_1 \to D$ be a measurable function. By definition of the conditional expectation we need to show for all $A_3 \in \mathcal{E}_3$ and all $A_{1,2} \in \mathcal{E}_1 \otimes \mathcal{E}_2$

$$\mathbb{E}\Big[\mathbf{1}_{A_{1,2}}(\xi_1, \xi_2^{(g(\xi_1))})\kappa_{3|2,1}\big(\xi_1, \xi_2^{(g(\xi_1))}, g(\xi_1); A_3\big)\Big] = \mathbb{E}\big[\mathbf{1}_{A_{1,2}}(\xi_1, \xi_2^{(g(\xi_1))})\mathbf{1}_{A_3}(\xi_3)\big]$$

Without loss of generality we may only consider $A_{1,2} = A_1 \times A_2 \in \mathcal{E}_1 \times \mathcal{E}_2$ since the product sigma algebra $\mathcal{E}_1 \otimes \mathcal{E}_2$ is generated by these rectangles. Since

$$\kappa_{2|1}^g(x_1; A_2) \coloneqq \kappa_{3,2|1}(x_1, g(x_1); E_3 \times A_2)$$

is a regular conditional version of $\mathbb{P}(\xi_2^{(g(\xi_1))} \in \cdot \mid \xi_1)$ by assumption (8) we may apply disintegration (e.g. Kallenberg, 2002, Thm. 6.4) to the measurable function

$$\varphi(x_1, x_2) \mapsto \mathbf{1}_{A_2}(x_2)\kappa_{3|2,1}(x_1, x_2, g(x_1); A_3)$$

to obtain

$$\mathbb{E}\big[\varphi\big(\xi_1, \xi_2^{g(\xi_1)}\big) \mid \xi_1\big] \overset{a.s.}{=} \int \varphi(\xi_1, x_2)\kappa_{2|1}^g(\xi_1; dx_2)$$

$$\overset{def.}{=} \int \varphi(\xi_1, x_2)\kappa_{3,2|1}(\xi_1, g(\xi_1); E_3 \times dx_2). \tag{10}$$

We thereby have

$$\mathbb{E}\Big[\mathbf{1}_{A_1}(\xi_1)\mathbf{1}_{A_2}(\xi_2^{g(\xi_1)})\kappa_{3|2,1}\big(\xi_1,\xi_2^{g(\xi_1)},g(\xi_1);A_3\big)\Big]$$

$$= \mathbb{E}\Big[\mathbf{1}_{A_1}(\xi_1)\varphi\big(\xi_1,\xi_2^{g(\xi_1)}\big)\Big]$$

$$\overset{(10)}{=} \mathbb{E}\Big[\mathbf{1}_{A_1}(\xi_1)\int \varphi(\xi_1,x_2)\kappa_{3,2|1}(\xi_1,g(\xi_1);E_3\times dx_2)\Big]$$

$$\overset{\text{Lemma } 4.4}{=} \mathbb{E}\Big[\mathbf{1}_{A_1}(\xi_1)\kappa_{3,2|1}(\xi_1,g(\xi_1);A_3\times A_2)\Big]$$

$$\overset{(8)}{=} \mathbb{E}\Big[\mathbf{1}_{A_1}(\xi_1)\mathbf{1}_{A_3\times A_2}(\xi_3,\xi_2^{g(\xi_1)})\Big]$$

$$= \mathbb{E}\Big[\mathbf{1}_{A_1}(\xi_1)\mathbf{1}_{A_2}(\xi_2^{g(\xi_1)})\mathbf{1}_{A_3}(\xi_3)\Big].$$

The crucial step is the application of Lemma 4.4, which provides an integral representation of a regular conditional distribution of $\xi_3,\xi_2^y\mid\xi_1$ that *couples* the two conditional kernels.

**Lemma 4.4.** *For all $A_2\in\mathcal{E}_2$, $A_3\in\mathcal{E}_3$, all $y\in\mathbb{X}$ and $\mathbb{P}_{\xi_1}$-almost all $x_1$*

$$\kappa_{3,2|1}(x_1,y;A_3\times A_2) = \int \varphi(x_1,x_2)\kappa_{3,2|1}(x_1,y;E_3\times dx_2) \tag{11}$$

$$= \int \mathbf{1}_{A_2}(x_2)\kappa_{3|2,1}(x_1,x_2,y;A_3)\kappa_{3,2|1}(x_1,y;E_3\times dx_2).$$

In the remainder of the proof we will show this Lemma. To this end pick any $A_1\in\mathcal{E}_1$. Then by definition of the conditional expectation (e.g. Klenke, 2014, chap. 8)

$$\mathbb{E}[\mathbf{1}_{A_1}(\xi_1)\kappa_{3,2|1}(x_1,y;A_3\times A_2)]$$

$$\overset{(8)}{=} \mathbb{E}[\mathbf{1}_{A_1}(\xi_1)\mathbf{1}_{A_2}(\xi_2^y)\mathbf{1}_{A_3}(\xi_3)]$$

$$\overset{(9)}{=} \mathbb{E}[\mathbf{1}_{A_1}(\xi_1)\mathbf{1}_{A_2}(\xi_2^y)\kappa_{3|2,1}(\xi_1,\xi_2^y;A_3)]$$

$$\overset{(*)}{=} \mathbb{E}\Big[\mathbf{1}_{A_1}(\xi_1)\int \mathbf{1}_{A_2}(x_2)\kappa_{3|2,1}(\xi_1,x_2,y;A_3)\kappa_{3,2|1}(\xi_1,y;E_3\times dx_2)\Big]$$

Note that the constant function $g\equiv y$ is always measurable for the application of (8). The last step $(*)$ is implied by disintegration (e.g. Kallenberg, 2002, Thm. 6.4)

$$\mathbb{E}[f(\xi_1,\xi_2^y)\mid\xi_1] \overset{\text{a.s.}}{=} \int f(\xi_1,x_2)\kappa_{2|1}^y(\xi_1;dx_2)$$

of the measurable function $f(x_1,x_2):=\mathbf{1}_{A_2}(x_2)\kappa_{3|2,1}(x_1,x_2;A_3)$ using the probability kernel

$$\kappa_{2|1}^y(x_1;A_2):=\kappa_{3,2|1}(x_1,y;E_3\times A_2)$$

which is a regular conditional version of $\mathbb{P}(\xi_2^y\in\cdot\mid\xi_1)$ by assumption (8), since the constant function $g\equiv y$ is measurable. $\square$

### 4.2.1 Proof of the main result (Theorem 2.10)

In this section we assume that $\mathbb{X}$ is a locally compact, separable and metrizable space and $\mathbb{Y}$ is a Polish space. In this setting, we can use Proposition 2.4 to get rid of the assumption in Proposition 4.2. This is done in the following corollary, which is then modified for conditional independence in Lemma 4.6. We finally prove our main result using repeated applications of this lemma.

**Corollary 4.5** (Automatic PIC)**.** *Let $Z$ be a random variable in a standard borel space $(E,\mathcal{B}(E))$, and $\mathbf{f}$ a random variable in $C(\mathbb{X},\mathbb{Y})$. Let $W$ be a random element in an arbitrary measurable space $(\Omega,\mathcal{F})$. If there exists a joint conditional distribution $\kappa$ for $Z$ given $(W,\mathbf{f}(x))$ then $\kappa$ is automatically PIC. That is, for all $B\in\mathcal{B}(E)$ all $\sigma(W)$ measurable $X$*

$$\mathbb{P}(Z\in B\mid W,\mathbf{f}(X)) \overset{a.s.}{=} \kappa(W,\mathbf{f}(X),X;B),$$

*Proof.* Since $X = g(W)$ for some measurable function $g$, Proposition 4.2 with $(\xi_3, \xi_2^y, \xi_1) = (Z, \mathbf{f}(y), W)$ yields the claim, since a PIC joint probability kernel for $\xi_3, \xi_2^y$ given $\xi_1$ *exists* by Proposition 2.4. $\qquad\square$

The proof of Theorem 2.10 (i) will follow from repeated applications of the following lemma. This lemma slightly generalizes Corollary 4.5 to the conditional independence setting. Theorem 2.10 (ii) will follow from (i) and an application of Proposition 2.4.

**Lemma 4.6** (PIC allows conditional independence)**.** *Let $Z$ be a random variable in a standard borel space $(E, \mathcal{B}(E))$, $\mathbf{f}$ a continuous random function in $C(\mathbb{X}, \mathbb{Y})$. Let $W$ be a random element in an arbitrary measurable space $(\Omega, \mathcal{F})$. If there exists a joint conditional distribution $\kappa$ for $Z$ given $W, \mathbf{f}(x)$ then for any random variable $X \perp\!\!\!\perp_W (Z, \mathbf{f})$ in $\mathbb{X}$*

$$\mathbb{P}(Z \in B \mid W, X, \mathbf{f}(X)) = \kappa(W, \mathbf{f}(X), X; B).$$

*Proof.* Observe that $X$ is clearly measurable with respect to $W^+ := (W, X)$. Our proof strategy therefore relies on constructing a joint conditional distribution for $Z$ given $(W^+, \mathbf{f}(x))$ using $\kappa$ and apply Corollary 4.5.

Since $X$ independent from $(Z, \mathbf{f})$ conditional on $W$ there exists a standard uniform $U \sim \mathcal{U}(0, 1)$ independent from $(W, Z, \mathbf{f})$ such that $X = h(W, U)$ for some measurable function $h$ (Kallenberg, 2002, Prop. 6.13). Since $U$ is independent from $W, Z, \mathbf{f}$ we have by (Kallenberg, 2002, Prop. 6.6)

$$\mathbb{P}(Z \in B \mid W, U, \mathbf{f}(x)) = \mathbb{P}(Z \in B \mid W, \mathbf{f}(x)) = \kappa(W, \mathbf{f}(x), x; B)$$

for all $x \in \mathbb{X}$. Since $\sigma(W, X, \mathbf{f}(x)) \subseteq \sigma(W, U, \mathbf{f}(x))$ and $(W, \mathbf{f}(x))$ is measurable with respect to $\sigma(W, X, \mathbf{f}(x))$ we therefore have

$$\mathbb{P}(Z \in B \mid \underbrace{W, X}_{=W^+}, \mathbf{f}(x)) = \kappa(W, \mathbf{f}(x), x; B) =: \kappa^+(\underbrace{W, X}_{=W^+}, \mathbf{f}(x), x; B),$$

where $\kappa^+$ is defined as constant in the second input. An application of Corollary 4.5 to $\kappa^+$ yields the claim. $\qquad\square$

*Proof of Theorem 2.10.* We will prove (i) by induction over $k \in \{0, \ldots, n+1\}$. For any conditionally independent evolution $(X_n)_{n \in \mathbb{N}_0}$, the induction claim is

$$\begin{aligned}
&\mathbb{P}\big(Z \in A \mid W, (\mathbf{f}_i(X_i))_{i \in [0:k)}, (\mathbf{f}_i(x_i))_{i \in [k:n]}, X_{[0:k)}\big) \\
&\qquad = \kappa\big(W, (\mathbf{f}_i(X_i))_{i \in [0:k)}, (\mathbf{f}_i(x_i))_{i \in [k:n]}, X_{[0:k)}, x_{[k:n]}; A\big).
\end{aligned} \tag{12}$$

That is, we plugged in all the random variables up to index $k-1$. The induction start with $k = 0$ is given by assumption and $k = n+1$ is the claim, so we only need to show the induction step $k \to k+1$. For this purpose we want to define $\tilde{W} = \big(W, (\mathbf{f}_i(X_i))_{i \in [0:k)}, (\mathbf{f}_i(x_i))_{i \in (k:n]}, X_{[0:k)}\big)$ and the kernel

$$\tilde{\kappa}_{x_{(k:n]}}\big(\tilde{W}, \mathbf{f}_k(x_k), x_k; A\big) := \kappa\big(W, (\mathbf{f}_i(X_i))_{i \in [0:k)}, (\mathbf{f}_i(x_i))_{i \in [k:n]}, X_{[0:k)}, x_{[k:n]}; A\big),$$

which is formally defined for any fixed $x_{(k:n]}$ by mapping the elements of $\tilde{W}$ into the right position. By induction (12) we thereby have

$$\mathbb{P}(Z \in A \mid \tilde{W}, \mathbf{f}(x_k)) = \tilde{\kappa}_{x_{(k:n]}}(\tilde{W}, \mathbf{f}(x_k), x_k; A).$$

We can now finish the induction using Lemma 4.6 if we can prove $X_k$ is independent from $(Z, \mathbf{f})$ conditional on $\tilde{W}$, because then we can also plug-in $X_k$. For the conditional independence we will use the characterization in Proposition 6.13 by Kallenberg (2002).

Since $X_k$ is independent from $(Z, \mathbf{f})$ conditional on $\mathcal{F}_{k-1}$ there exists, by this Proposition, a uniform random variable $U \sim \mathcal{U}(0, 1)$ independent from $(Z, \mathbf{f}, \mathcal{F}_{k-1})$ such that $X_k = h(\xi, U)$ for some measurable function $h$ and a random element $\xi$ that generates $\mathcal{F}_{k-1}$. Due to $\mathcal{F}_{k-1} \subseteq \sigma(\tilde{W})$ the element $\xi$ is a measurable function of $\tilde{W}$ and therefore $X_k = \tilde{h}(\tilde{W}, U)$ for some measurable function $\tilde{h}$. Since $U$ is independent from $(Z, \mathbf{f}, \mathcal{F}_{k-1})$,

it is independent from $\tilde{W}$ as $\sigma(\tilde{W}) \subseteq \sigma(\mathbf{f}, \mathcal{F}_{k-1})$. Using Prop. 6.13 from Kallenberg (2002) again, $X_k$ is thereby independent from $(Z, \mathbf{f})$ conditional on $\tilde{W}$.

What remains is the proof of (ii). Let $x_n$ be fixed and define $\tilde{Z} = (Z, \mathbf{f}_n(x_n))$. Since $\kappa$ is a joint conditional distribution for $Z, \mathbf{f}_n(x_n)$ given $\mathcal{F}, (\mathbf{f}_k(x_k))_{k \in [0:n)}$ the kernel

$$\kappa_{x_n}(y_{[0:n)}, x_{[0:n)}; B) := \kappa(y_{[0:n)}, x_{[0:n]}; B)$$

clearly satisfies the requirements of (i) and thereby

$$\mathbb{P}\big(Z, \mathbf{f}(x_n) \in B \mid \mathcal{F}_{n-1}\big) = \kappa\big(\underbrace{W, (\mathbf{f}_k(X_k))_{k \in [0:n)}, X_{[0:n)}}_{=: \tilde{W}}, x_n; B\big) =: \tilde{\kappa}(\tilde{W}, x_n; B).$$

Since $\tilde{W}$ generates $\mathcal{F}_{n-1}$ we are almost in the setting of Proposition 2.4, as we have continuity in $x_n$. However, since $X_n$ is not previsible we have to repeat the same trick we used in the proof of Lemma 4.6. Namely, $X_n$ is measurable with respect to $W^+ := (\tilde{W}, X_n)$ and we will have to construct a joint conditional distribution for $Z, \mathbf{f}(x_n)$ given $W^+$.

Since $X_n$ is independent from $Z, \mathbf{f}$ conditional on $\tilde{W}$, there exists a standard uniform $U \sim \mathcal{U}(0, 1)$ independent from $(\tilde{W}, Z, \mathbf{f})$ such that $X_n = h(\tilde{W}, U)$ for some measurable function $h$ (Kallenberg, 2002, Prop. 6.13). Since $U$ is independent from $(\tilde{W}, Z, \mathbf{f})$, we have by (Kallenberg, 2002, Prop. 6.6)

$$\mathbb{P}(Z, \mathbf{f}(x_n) \in B \mid \tilde{W}, U) \overset{\text{a.s.}}{=} \mathbb{P}(Z, \mathbf{f}(x_n) \in B \mid \tilde{W}) \overset{\text{a.s.}}{=} \tilde{\kappa}(\tilde{W}, x_n; B)$$

for all $x_n \in \mathbb{X}$. Since $\sigma(\tilde{W}, X_n) \subseteq \sigma(\tilde{W}, U)$ and $\tilde{W}$ is measurable with respect to $\sigma(\tilde{W}, X_n)$ we therefore have

$$\mathbb{P}(Z, \mathbf{f}(x_n) \in B \mid \underbrace{\tilde{W}, X_n}_{=W^+}) \overset{\text{a.s.}}{=} \tilde{\kappa}(\tilde{W}, x; B) =: \kappa^+(\underbrace{\tilde{W}, X_n}_{=W^+}, x_n; B),$$

where $\kappa^+$ is defined as constant in the second input. Clearly, by definition of $\kappa^+$ via $\kappa$, $\kappa^+$ is continuous in $x_n$ and as a continuous joint conditional distribution it is PIC by Proposition 2.4. This finally implies the claim

$$\mathbb{P}(Z, \mathbf{f}(X_n) \in B \mid \underbrace{\mathcal{F}_{n-1}, X_n}_{=W^+}) \overset{\text{a.s.}}{=} \kappa^+(W^+, X_n; B)$$
$$= \kappa(W, (\mathbf{f}_k(X_k))_{k \in [0:n)}, X_{[0:n)}, X_n; B). \qquad \square$$

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

## A  Gaussian conditionals

In this section we want to recall the canonical version of the conditional distribution of Gaussian random vectors/functions.

*Remark* A.1 (Multivariate output). While we limit ourselves to real-valued functions in the following lemma, the result may easily be extended to $\mathbb{R}^d$ valued functions. To do so simply turn the output dimension into an input, i.e. $\mathbf{f}(x)_i =: \mathbf{f}(x, i)$ for $i \in [1{:}d]$, where we recall the notation for discrete intervals

$$[i{:}j] := [i, j] \cap \mathbb{Z}, \qquad [i{:}j) := [i, j) \cap \mathbb{Z}, \qquad \text{etc.}$$

In other words we simply replace the domain $\mathbb{X}$ by $\mathbb{X} \times [1{:}d]$. This necessitates that we do not only consider $\mathbf{f}(x_n)$ given $\mathbf{f}(x_0), \ldots, \mathbf{f}(x_{n-1})$ but instead the the distribution of $\mathbf{f}(x_k), \ldots, \mathbf{f}(x_n)$ given $\mathbf{f}(x_0), \ldots, \mathbf{f}(x_{k-1})$.

**Lemma A.2** (Gaussian conditional distribution). *Let $(\mathbf{f}(x))_{x \in \mathbb{X}}$ with values in $\mathbb{R}$ be a Gaussian random function with mean and covariance function*

$$\mu(x) = \mathbb{E}[\mathbf{f}(x)] \qquad and \qquad \mathcal{C}_{\mathbf{f}}(x, y) = \mathrm{Cov}(\mathbf{f}(x), \mathbf{f}(y)).$$

*Then a regular conditional distribution*

$$\mathbb{P}\Big( \big( \mathbf{f}(x_{k+1}), \ldots, \mathbf{f}(x_n) \big) \in A \mid Y \Big) = \kappa_x(Y; A)$$

*for $k < n$, $Y = (\mathbf{f}(x_0), \dots, \mathbf{f}(x_k))$ and $x = (x_0, \dots, x_n)$ is given by the Gaussian distribution*

$$\kappa_x(y; A) = \kappa(x, y; A)$$
$$\propto \int_A \exp\Big(-\tfrac{1}{2}\big(t - \mu_{n|k}(x, y)\big)^T [\Sigma_{n|k}(x)]^{-1}\big(t - \mu_{n|k}(x, y)\big)\Big) dt, \tag{13}$$

*where the posterior mean is given by*

$$\mu_{n|k}(x, y) = \begin{pmatrix} \mu(x_{k+1}) \\ \vdots \\ \mu(x_n) \end{pmatrix} + M\Big[(\mathcal{C}_\mathbf{f}(x_i, x_j))_{i,j \in [0:k]}\Big]^{-1}\Big(y - \begin{pmatrix} \mu(x_0) \\ \vdots \\ \mu(x_k) \end{pmatrix}\Big)$$

*and posterior covariance by*

$$\Sigma_{n|k}(x) = (\mathcal{C}_\mathbf{f}(x_i, x_j))_{i,j \in (k:n]} - M\Big[(\mathcal{C}_\mathbf{f}(x_i, x_j))_{i,j \in [0:k]}\Big]^{-1} M^T$$

*with*

$$M = \begin{pmatrix} \mathcal{C}_\mathbf{f}(x_{k+1}, x_0) & \cdots & \mathcal{C}_\mathbf{f}(x_{k+1}, x_k) \\ \vdots & & \vdots \\ \mathcal{C}_\mathbf{f}(x_n, x_0) & \cdots & \mathcal{C}_\mathbf{f}(x_n, x_k) \end{pmatrix}.$$

*Proof.* Since $(\mathbf{f}(x_0), \dots, \mathbf{f}(x_n))$ is multivariate Gaussian, this is simply a standard result about conditionals of Gaussian vectors (e.g. Eaton, 2007, Prop. 3.13). □

**Definition A.3** (Canonical Gaussian conditional distribution)**.** Since the definition of the regular conditional distribution in (13) is not unique, we refer to this specific version as the *canonical* Gaussian conditional distribution.

*Remark* A.4 (Joint kernel)*.* The collection of kernels $(\kappa_x)_{x \in \mathbb{X}^{n+1}}$ is a joint distribution, since it is measurable in $x, y$ for fixed $A$.

*Remark* A.5 (Continuity)*.* If $\mu$ and $\mathcal{C}_\mathbf{f}$ are continuous, then the characteristic function of the joint kernel

$$\hat{\kappa}(x, y; u) = \exp\Big(iu^T \mu_{n|k}(x, y) - \tfrac{1}{2} u^T \Sigma_{n|k}(x) u\Big)$$

is continuous in $x_{(k:n]}$, since both $\mu_{n|k}$ and $\Sigma_{n|k}$ are continuous in $x_{(k:n]}$. This can be seen directly from the explicit expressions in Lemma A.2 as the $x_j$ with $j > k$ are not involved in the matrix inversion. This implies continuity of $x_{(k:n]} \mapsto \kappa(x, y; \cdot)$ in the weak topology by Lévy's continuity theorem (e.g. Kallenberg, 2002, Thm. 5.3).

*Remark* A.6 (Artificially breaking things)*.* This canonical kernel may be artificially modified to be discontinuous. E.g. with an indicator on $\{x = y\}$, which is a null set for every fixed $x$ as $Y$ has a density. The resulting collection of kernels would still be a joint regular conditional distribution, however in contrast to the canonical Gaussian conditional distribution they would not necessarily be PIC (see Prop. 2.4). Similarly, the collection may be modified to not be a joint kernel.

## B  Topological foundation

In this section we show the evaluation function to be continuous and therefore measurable for continuous random functions. For *compact* $\mathbb{X}$ this result can be collected from various sources (e.g. Engelking, 1989, Thm. 4.2.17 and Kechris, 1995, Thm. 4.19). But we could not find a reference for the result in this generality, so we provide a proof.

**Theorem B.1** (Continuous functions)**.** *Let $\mathbb{X}$ be a locally compact, separable and metrizable space[7], $\mathbb{Y}$ a Polish space and $C(\mathbb{X}, \mathbb{Y})$ the space of continuous functions equipped with the* compact-open[8] *topology. Then*

---

[7]We do not need $\mathbb{X}$ to be metrizable but only regular and second countable (in metrizable spaces 'second countable' is equivalent to 'separable' (Engelking, 1989, Cor. 4.1.16)). We choose this more specific definition to make it more obvious that a locally compact Polish space satisfies the requirements. However, for the proof we will assume the more general setting.

[8]The sets $M(K, U) := \{f \in C(\mathbb{X}, \mathbb{Y}) : f(K) \subseteq U\}$ with $K \subseteq \mathbb{X}$ compact and $U \subseteq \mathbb{Y}$ open, form a sub-base of the *compact-open* topology (e.g. Engelking, 1989, Sec. 3.4). I.e. the compact-open topology it is the smallest topology such that all $M(K, U)$ are

(i) *the evaluation function*

$$e \colon \begin{cases} C(\mathbb{X}, \mathbb{Y}) \times \mathbb{X} \to \mathbb{Y} \\ (f, x) \mapsto f(x) \end{cases}$$

*is continuous and therefore measurable.*

(ii) $C(\mathbb{X}, \mathbb{Y})$ *is a* **Polish space**, *whose topology is generated by the metric*

$$d(f, g) \coloneqq \sum_{n=1}^{\infty} 2^{-n} \frac{d_n(f, g)}{1 + d_n(f, g)} \quad \text{with} \quad d_n(f, g) \coloneqq \sup_{x \in K_n} d_{\mathbb{Y}}(f(x), g(x))$$

*for any metric $d_{\mathbb{Y}}$ that generates the topology of $\mathbb{Y}$ and any* compact exhaustion[9] $(K_n)_{n \in \mathbb{N}}$ *of $\mathbb{X}$, that always exists because $\mathbb{X}$ is hemicompact!*

(iii) *The Borel $\sigma$-algebra of $C(\mathbb{X}, \mathbb{Y})$ is equal to the restriction of the product sigma algebra of $\mathbb{Y}^{\mathbb{X}}$ to $C(\mathbb{X}, \mathbb{Y})$, i.e. $\mathcal{B}(C(\mathbb{X}, \mathbb{Y})) = \mathcal{B}(\mathbb{Y})^{\otimes \mathbb{X}}\big|_{C(\mathbb{X}, \mathbb{Y})}$.*

*Remark* B.2 (Topology of pointwise convergence). The topology of point-wise convergence ensures that all projection mappings $\pi_x(f) = f(x)$ are continuous. It coincides with the product topology (Engelking, 1989, Prop. 2.6.3). Thm. B.1 (iii) ensures that the Borel-$\sigma$-algebra generated by the topology of point-wise convergence coincides with the Borel $\sigma$-algebra generated by the compact-open topology.

*Remark* B.3 (Construction). The main tool for the construction of probability measures, Kolmogorov's extension theorem (e.g. Klenke, 2014, Sec. 14.3), allows for the construction of random measures on product spaces. This is only compatible with the product topology, i.e. the topology of point-wise convergence. But the evaluation map is generally not continuous with respect to this topology (Engelking, 1989, Prop. 2.6.11). (iii) ensures that this does not pose a problem as long as $\mathbb{X}$ and $\mathbb{Y}$ satisfy the requirements of Theorem B.1 and the constructed random process has a continuous version (cf. Talagrand, 1987, Thm. 3, Costa et al., 2024 and references therein).

*Remark* B.4 (Limitations). While the compact-open topology can be defined for general topological spaces, the continuity of the evaluation map crucially depends on $\mathbb{X}$ being locally compact (Engelking, 1989, Thm. 3.4.3 and comments below). For $\mathbb{X}$ and $\mathbb{Y}$ Polish spaces, this implies $C(\mathbb{X}, \mathbb{Y})$ is generally only well behaved if $\mathbb{X}$ is locally compact.

*Remark* B.5 (Discontinuous case). Without continuity it is already difficult to obtain a random function $\mathbf{f}$ that is almost surely measurable and can be evaluated point-wise. The construction of Lévy processes in càdlàg[10] space only works on ordered domains such as $\mathbb{R}$, where 'right-continuous' has meaning. Typically, discontinuous random functions are therefore only constructed as generalized functions in the sense of distributions[11] that cannot be evaluated point-wise (e.g. Schäffler, 2018). In particular, we cannot hope to evaluate generalized random functions at random locations. Nevertheless it may be possible to prove the evaluation function to be measurable for more general separable functions.

*Proof.* Since $\mathbb{X}$ is locally compact, (i) follows from Proposition 2.6.11 and Theorem 3.4.3. by Engelking (1989).

For (ii) let us begin to show that $\mathbb{X}$ is **hemicompact/exhaustible by compact sets**. Since the space $\mathbb{X}$ is locally compact, pick a compact neighborhood for every point. The interiors of these compact neighborhoods obviously cover $\mathbb{X}$. Since every regular, second countable space[7] is Lindelöf (Engelking, 1989, Thm. 3.8.1), we can pick a countable subcover, such that the interiors of the sequence $(C_i)_{i \in \mathbb{N}}$ of compact sets cover the domain $\mathbb{X}$. We inductively define a compact exhaustion $(K_n)_{n \in \mathbb{N}}$ with $K_1 \coloneqq C_1$. Observe that the set $K_n$

---

open. Recall that the set of finite intersections of a sub-base form a base of the topology and elements from the topology can be expressed as unions of base elements.

[9] The set $\mathbb{X}$ is *hemicompact* if it can be *exhausted by the compact sets* $(K_n)_{n \in \mathbb{N}}$, which means that the compact set $K_n$ is contained in the interior of $K_{n+1}$ for any $n$ and $\mathbb{X} = \bigcup_{n \in \mathbb{N}} K_n$.

[10] french: continue à droite, limite à gauche, "right-continuous with left-limits"

[11] The set of distributions is defined as the topological dual to a set of test functions. In particular, distributions are *continuous* linear functionals acting on the test functions. Thereby one may hope that Theorem B.1 is applicable, but the set of test functions is typically not locally compact (cf. Remark B.4).

is covered by the interiors $(\text{int}\, C_i)_{i\in\mathbb{N}}$. Since $K_n$ is compact, we can choose a finite sub-cover $(\text{int}\, C_i)_{i\in I}$ and define $K_{n+1} := \bigcup_{i\in I} C_i \cup C_{n+1}$. Then by definition $K_n$ is contained in the interior of the compact set $K_{n+1}$ and due to $C_n \subseteq K_n$ this sequence also covers the space $\mathbb{X}$ and is thereby a compact exhaustion.

It is straightforward to check that the metric defined in (ii) is a metric, so we will only prove this **metric induces the compact-open** topology.

(I) **The compact-open topology is a subset of the metric topology.** We need to show that the sets $M(K,U)$ are open with respect to the metric. This requires for any $f \in M(K,U)$ an $\epsilon > 0$ such that the epsilon ball $B_\epsilon(f)$ is contained in $M(K,U)$.

We start by constructing a finite cover of $f(K)$. For any $x \in K$ there exists $\delta_x > 0$ with $B_{2\delta_x}(f(x)) \subseteq U$ for balls induced by the metric $d_\mathbb{Y}$ as $U$ is open. Since $K$ is compact, $f(K) \subseteq U$ is a compact set covered by the balls $B_{\delta_x}(f(x))$. This yields a finite subcover $B_{\delta_1}(f(x_1)), \ldots, B_{\delta_m}(f(x_m))$ of $f(K)$.

Using this cover we will prove the following criterion: Any $g \in C(\mathbb{X}, \mathbb{Y})$ is in $M(K,U)$ if

$$\sup_{x\in K} d_\mathbb{Y}(f(x), g(x)) < \delta := \min\{\delta_1, \ldots, \delta_m\}. \tag{14}$$

For this criterion note that for any $x \in K$ there exists $i \in \{1, \ldots, m\}$ such that $f(x) \in B_{\delta_i}(f(x_i))$. This implies

$$d(g(x), f(x_i)) \le d(g(x), f(x)) + d(f(x), f(x_i)) \le 2\delta_i,$$

which implies $g(K) \subseteq \bigcup_{i=1}^m B_{2\delta_i}(f(x_i)) \subseteq U$ and therefore $g \in M(K,U)$.

Consequently, if there exists $\epsilon > 0$ such that $g \in B_\epsilon(f)$ implies criterion (14), then we have $B_\epsilon(f) \subseteq M(K,U)$ which finishes the proof. And this is what we will show. Since $K$ is compact and the interiors of $K_n$ cover the space, there exists a finite sub-cover $K \subseteq \bigcup_{i\in I} K_i$ and therefore some $m = \max I$ such that $K$ is in the interior of $K_m$. By definition of $d_m$ it is thus clearly sufficient to ensure $d_m(f,g) < \delta$. And since $\varphi(x) = \frac{x}{1+x}$ is a strict monotonous function $\epsilon := 2^{-m}\varphi(\delta)$ does the job, since $2^{-m}\varphi(d_m(f,g)) \le d(f,g) \le \epsilon$ implies $d_m(f,g) \le \delta$.

(II) **The metric topology is a subset of the compact-open topology.** Since the balls $B_\epsilon(f)$ form a base of the metric topology it is sufficient to prove them open in the compact-open topology. If for any $g \in B_\epsilon(f)$ there exists a compact $C_1, \ldots, C_m \subseteq \mathbb{X}$ and open $U_1, \ldots, U_m \subseteq \mathbb{Y}$ such that $g \in \bigcap_{j=1}^m M(C_j, U_j) \subseteq B_\epsilon(f)$, then the ball is open since these finite intersections are open sets in the compact-open topology and their union over $g$ remains open. But since there exists $r > 0$ such that $B_r(g) \subseteq B_\epsilon(f)$, it is sufficient to prove for any $r > 0$ that there exist compact $C_j$ and open $U_j$ such that

$$g \in V := \bigcap_{j=1}^m M(C_j, U_j) \subseteq B_r(g). \tag{15}$$

For this purpose pick $K_N$ from the compact exhaustion with sufficiently large $N$ such that $2^{-N} < \frac{r}{2}$. Pick a finite cover $O_{x_1}, \ldots O_{x_m}$ of $K_N$ from the cover $\{O_x\}_{x\in K_N}$ with $O_x := g^{-1}(B_{r/5}(g(x)))$ and define the sets

$$C_j := \overline{O_{x_j}} \cap K_N \qquad U_j := B_{r/4}(f(x_j)).$$

Clearly the $U_j$ are open and the $C_j$ are compact and we will now prove they satisfy (15). Observe that $g \in V$ since for all $j$

$$g(C_j) \subseteq g(\overline{O_{x_j}}) \subseteq \overline{B_{r/5}(f(x_j))} \subseteq U_j.$$

Pick any other $h \in V$. Then for all $x \in K_N$ there exists $i$ such that $x \in O_{x_i} \subseteq C_i$ and by definition of $V$ this implies $h(x) \in U_i$ and also $g(x) \in U_i$ and thereby $d_\mathbb{Y}(h(x), g(x)) \le r/2$. This uniform bound implies $d_N(h,g) \le r/2$ and therefore

$$d(g,h) \le \left(\sum_{n=1}^N 2^{-n} d_n(g,h)\right) + \left(\sum_{n=N+1}^\infty 2^{-n}\right) \le d_N(g,h) + 2^{-N} < r,$$

since $d_n(f,g) \le d_N(f,g)$ for $n \le N$. Thus $h \in B_r(g)$ which proves (15).

As $C(\mathbb{X}, \mathbb{Y})$ is clearly metrizable, what is left to prove are its separability and completeness. Separability could be proven directly similarly to the proof of Theorem 4.19 in Kechris (1995) but for the sake of brevity this result follows from Theorem 3.4.16 and Theorem 4.1.15 (vii) by Engelking (1989) and the fact that $\mathbb{X}$ and $\mathbb{Y}$ are second countable. Completeness follows from the fact that any Cauchy sequence $f_n$ induces a Cauchy sequence $f_n(x)$ for any $x$ by definition of the metric. And by completeness of $\mathbb{Y}$ there must exist a limiting value $f(x)$ for any $x$. The continuity of $f$ follows from the uniform convergence on compact sets, since every compact set is contained in some $K_n$ from the compact exhaustion (cf. last paragraph in (I)).

What is left to prove is (iii). Since the projections are continuous with respect to the compact open topology, they are measurable with respect to the Borel-$\sigma$-algebra. The product sigma algebra, which is the smallest sigma algebra to ensure all projections are measuralbe, restricted to the continuous functions is therefore a subset of the Borel $\sigma$-algebra. To prove the opposite inclusion, we need to show that the open sets are contained in the product $\sigma$-algebra. Since the space is second countable (Engelking, 1989, Cor. 4.1.16) and every open set thereby a countable union of its base, it is sufficient to check that the open ball $B_\epsilon(f_0)$ for $\epsilon > 0$ and $f_0 \in C(\mathbb{X}, \mathbb{Y})$ is in the product sigma algebra restricted to $C(\mathbb{X}, \mathbb{Y})$. But since $B_\epsilon(f_0) = H^{-1}([0, \epsilon))$ with $H(f) := d(f, f_0)$, it is sufficient to prove $H$ is $\sigma(\pi_x : x \in \mathbb{X})$-$\mathcal{B}(\mathbb{R})$-measurable, where $\pi_x$ are the projections. $H$ is measurable if $H_n(f) = d_n(f, f_0)$ is measurable, as a limit, sum, etc. (Klenke, 2014, Thm. 1.88-1.92) of measurable functions. But since $\mathbb{X}$ is separable (Engelking, 1989, Cor. 1.3.8), i.e. has a countable dense subset $Q$, we have by continuity of $f$ and $f_0$

$$d_n(f, f_0) = \sup_{x \in K_n} d_{\mathbb{Y}}(f(x), f_0(x)) = \sup_{x \in K_n \cap Q} d_{\mathbb{Y}}(\pi_x(f), \pi_x(f_0)).$$

Since $d_{\mathbb{Y}}$ is continuous and thereby measurable (Klenke, 2014, Thm. 1.88), $H_n$ is measurable as a countable supremum of measurable functions (Klenke, 2014, Thm. 1.92). □

