# OpenReview forum: "Measure Theory of Conditionally Independent Random Function Evaluation"
_TMLR — Rejected by TMLR_

### Review · Reviewer_vYVd · 2026-05-10

**Summary Of Contributions:**

This paper is a purely theoretical paper, that investigates the measure-theoretic validity of treating the adaptively chosen evaluation locations as "deterministic" when plugging them back into the conditional distributions of a random function. The paper defines "plug-in consistency" as the case when this is valid, and shows, through a counterexample, that in general this is not the case, but provides positive examples through continuous functions.

**Audience:**

Yes

**Audience Explanation:**

I, for one, value these papers very highly. Even though such foundational papers may appear to have no immediate practical benefit, it enhances the understanding of the community of a particular problem, and can in the long term have material benefits. This particular paper investigates the validity of a commonplace heuristic in Bayesian optimisation (which I must admit I am not familiar with), and I am convinced that the field will be stronger as a result of more solid theoretical foundations as provided by papers such as these.

**Broader Impact Concerns:**

The work is purely theoretical, so I do not see any broader impact concerns.

**Claims And Evidence:**

No

**Claims Explanation:**

The main theorem, Theorem 2.10(i), appears to be false. In fact, it seems to use an argument which, to my understanding, the entire point of this paper is telling us not to use. The hypothesis is that we have $\kappa$ being the conditional distribution of $Z$ given $\mathcal{F},\mathbf{f}_i(x_i)$ for *each fixed* $x_i$, and the conclusion is that we have PIC for *variables*. Counterexample 2.13 says we cannot do this in general. It appears we cannot do this even without dependent variables, contrary to the claim in Theorem 2.10(i).

Specifically, Lemma 4.4 appears to be correct, but is stated and proved for *each fixed $y$*. But then it is plugged into the proof of Proposition 4.2 as a random element $g(\xi_1)$. Lemma 4.4 only holds $\mathbb{P}_{\xi_1}$-almost surely for each $y$, with the null set depending on $y$, so $y$ cannot be plugged in with dependence on $\xi_1$.

Also, a fixable (but rather glaring) mistake in Counterexample 2.13 is that $Y$ is normally distributed with mean 0, but a few lines later it is written $0\neq\mathbb{E}[Y]$.

It is possible there are mathematical errors I didn't spot, so I would recommend the authors to carefully review their work, as the authors clearly have an extremely strong technical background (probably stronger than myself).

**Requested Changes:**

Theorem 2.10(i) needs to be changed substantially. I think you can add a counterexample to the statement of Theorem 2.10(i) as written. If you add a continuity hypothesis it will be subsumed by (ii) I imagine.

---

> ### Author Response · Authors · 2026-05-13
>
> Dear Reviewer,
>
> thank you for this thorough review and the identification for the mistake in the proof of Theorem 2.10(i). After some deliberation we believe you are unfortunately correct. Our only excuse is that we expected this to be easier to prove in every step of the way since it is so common to implicitly use these results in the calculation of conditional distributions. We now agree that not only the proof is wrong, the statement is also wrong. And unfortunately this is not easy to fix with a continuity assumption for two reasons:
> 1. Continuity in the conditional part cannot necessarily be expected even in the Gaussian case because it involves the inverse of covariance matrices. And if two points are equal, then the covariance matrix becomes degenerate. So it is impossible to prove continuity as soon as there are 2 points involved. One could try to work around this with separability (without continuity) but this does not fix the next point
> 2. The reason continuity was helpful for Proposition 2.4 (the dependent case), was that we could identify the existence of a continuous PIC kernel and then continuity of a different joint kernel is enough to prove that these two kernels are equal. This is the method we use to prove that any continuous joint kernel is PIC. However for the conditional side we could not prove the existence of a PIC kernel and therefore we cannot use this PIC kernel as an anchor to make the argument that some other kernel we can actually construct explicitly is PIC.
>
> Note that the distribution $(f(x_1),..., f(x_n))$ for a Gaussian random function $f$ is only known when the $x_1,...,x_n$ are deterministic. We thought it is enough to construct a joint kernel for all possible selections of $x_1,...,x_n$ and get a PIC kernel for free as there is no other known way to construct the conditional distribution when the inputs are random variables. As this appears to be wrong it is again unclear how to justify this extremely common practice. That is: It is unclear how to calculate conditional distributions of Gaussian random functions with random input, which is what all Bayes arguments are built on.
>
> In some earlier iterations we tried to move the result for the dependent case back to the conditonal part using Bayes arguments on densities. We will consider if these can be made to work. Although the justification of densities is very problematic even in the Gaussian case for the same reason that continuity is problematic: Covariance matrices can be degenerate. The second reason we discarded the density argument is that they are again only well defined up to null sets which turns into its own kind of nightmare. We are also vaguely aware of a different way to define conditional probabilities using nets that may be more fruitful.
>
> In summary we must admit that it is unclear how long it will take to find a suitable replacement argument. Thank you again for identifying this glaring issue.

---

> > ### Comment · Reviewer_vYVd · 2026-05-18
> >
> > Dear authors,
> >
> > Thank you very much for your honest and detailed reply, I really appreciate it.
> >
> > As I said in my review, I personally value such works as these very much, even though they may not gain immediate traction by the community. I hope you can find a good replacement argument.
> >
> > Reviewer.

---

### Review · Reviewer_TJN4 · 2026-05-21

**Summary Of Contributions:**

The paper studies the following measure-theoretic setup: Assume  $f$ is a random function and $\mathcal{F}$ a sigma algebra. Then there is generally a regular conditional distribution $\kappa_x$ such that $P(f(x)\in A|\mathcal{F})(\omega)=\kappa_x(\omega,A)$. However, this does not imply that $P(f(X)\in A|\mathcal{F})(\omega)=\kappa_X(\omega,A)$ for all $\mathcal{F}$ measurable random variables $X$. The paper then investigates what conditions on $\kappa$ ensure that this holds (plug in consistency) and shows existence of such $\kappa$. The main results extends this setup to cases where $X$ is not $\mathcal{F}$ measurable but a conditional independence relation holds.

Strength:

The paper resolves a subtle but existing problem. It relies on rigorous math and appears to be mostly solid.

Weaknesses:

See below, I am not sure whether the current presentation aligns with the TMLR audience. There are a few questions about the proofs.

**Additional Comments:**

Some questions and minor remarks:

- why do you often only condition on $f(X_i)$, don't you want to condition on $(f(X_i),X_i)$?
- p2.: "the evaluation map is only well known..." -> ... is only known ...
- "that generates $\mathcal{F}_n$", potentially write s.t. $\sigma(\xi)=\mathcal{F}_n$.
- In Theorem 2.10: Do you expect that existence generally holds beyond the Gaussian case? Have you tried to come up with counterexamples?
- In Counterexample 2.13: The mean of $Y$ should not be 0.
- But $f(X)$ is not necessarily Gaussian if $X$ is random: Why should it be? Isn't it  a mixture of Gaussian distributions?
- p. 7: "The goal is to find a maximum" local or global?
- In Example 3.3: What is the meaning of the last equation? The object is a random variable, what does it mean to maximize the RV over $x_n$?
- borel -> Borel
- When you refer to results from the literature, it might sometimes help to say in words what the result says so the manipulations are easier to follow.
- First eq. on p. 11: Explain how exactly disintegration is applied to obtain the given statement.
- Section 4.2: Provide a motivation for the setting (explain how it is used later) and provide an intuition why the result holds.
- In Proposition 4.2: Why isn't a similar diagonal counterexample possible? E.g., let $\xi_3=0$ and $\xi_2^y=0$, and $\xi_1$ uniform on $[0,1]$. For $\kappa_{3,2|1}$ outputting always $(0,0)$ should be PIC. If $\kappa_{3|1,2}(\xi_1,\xi_2^y,y;\cdot)$ is always $\delta_0$ (Dirac at 0) except when $y=\xi_1$. But then $g(\xi_1)=\xi_1$ should be problematic. Maybe the $y$ dependence should not be in the kernel but similar constructions seem possible. What am I missing?
- For the Gaussian conditionals state that you assume non-degenerate covariance matrices?

**Audience:**

No

**Audience Explanation:**

The problem is relevant to people from the ML community. However, I am not sure whether there is really a subset of the TMLR audience interested in the paper. One could argue that the paper is technical, requires substantial measure theoretic prerequisites, and its findings (only) justify that a commonly used practice is indeed valid. It seems that a statistic/probability journal might be a better fit for the paper. For the TMLR audience it might make sense to make the paper easier to digest. E.g., a less artificial counterexample which could occur in practice might make the paper more convincing, more proof heuristics can make it easier to follow, explain results from the literature a bit more to make the paper understandable to readers who are not so familiar with measure theory.

**Claims And Evidence:**

Yes

**Claims Explanation:**

The statements are rigorous and full proofs are provided. The results seem plausible (in particular the first one), however, I did not manage to verify the proofs (see also questions below). Providing a bit more intuition might make the arguments more transparent.

**Requested Changes:**

I made a few suggestions above. Overall, my main concern is a lack of fit, so making the paper a bit more accessible might be helpful for the TMLR audience.
Furthermore, depending on the answers to the questions below, further adjustments might be required.

---

> ### Author Response · Authors · 2026-05-28
>
> Thank you for your careful review and your many constructive suggestions for improvements. Indeed Proposition 4.2 is wrong as it is stated and we explained why this also breaks downstream results in a detailed answer to reviewer vYVd. Unfortunately we do not know of a way to fix this at this moment.

---

### Review · Reviewer_9Aw1 · 2026-05-25

**Summary Of Contributions:**

The paper formalises a common heuristic in Bayesian optimisation (BO) and related areas, specifically treating adaptively chosen evaluation locations $X_n$ as deterministic when computing conditional distributions of a random function $\mathbf{f}$. The key idea introduced is plug-in consistency (PIC) of joint kernels (the precise condition needed for the heuristic to hold). The main positive results are that PIC holds for continuous random functions (via a pushforward of the regular conditional distribution of $(Z,\mathbf{f})\mid\mathcal{F}$), an extension to the conditionally independent evolutions setting, and a Gaussian corollary that covers BO applications. The motivation is good and fills a (real) measure-theoretic gap that is routinely glossed in the BO and related literature. Further strengths being the clear presentation of the paper and it is overall well-written. The main weakness is that perhaps some of the claims may need revising for correctness.

**Audience:**

Yes

**Audience Explanation:**

A rigorous criterion for treating adaptive evaluation locations as deterministic in Gaussian conditional formulas would fill a real measure-theoretic gap (one that may be glossed over in Bayesian optimisation and the adjacent literature). I think readers in those communities (and perhaps those interested in measure-theoretic foundations of adaptive design more broadly) should find both the PIC framing and the corrected positive results interesting and useful.

**Claims And Evidence:**

No

**Claims Explanation:**

The main issue, at least as I understand things, is that Proposition 4.2 ("consistency shuffle") is false as stated. The proof of Prop. 4.2 applies Lemma 4.4, a fixed-$y$ conditional identity, with a $y$-dependent null set, at the random value $y=g(\xi_1)$. This seems like  the diagonal/null-set move the paper's Counterexample 2.13 is meant to warn against, with per-$y$ null sets invisible at any deterministic $y$ but can union into something of full measure along the random diagonal. A short example: $\xi_1$ uniform on $[0,1]$, $\xi_2^y \equiv 0$, and $\xi_3$ Bernoulli$(1/2)$ independent of $\xi_1$; with the outer kernel trivially PIC and inner kernel $\kappa_{3|2,1}(x_1, 0, y; \{1\}) = 1_{\{x_1 = y\}} + \frac{1}{2} 1_{\{x_1 \ne y\}}$, the proposition's hypotheses hold but at $g(x_1) = x_1$ the kernel returns $1$ while the true conditional is $1/2$. The same construction with $W = \xi_1$, $Z = \xi_3$, $f \equiv 0$, $X = W$ breaks Corollary 4.5 ("Automatic PIC") and Theorem 2.10(i) directly.

Counterexample 2.13 seems to be arithmetically incorrect, with $Y \sim N(0,1)$, $E[Y] = 0$, so $g(x) = E[Y] \, 1_{\{U \ne x\}} \equiv 0$ and the displayed inequality collapses to $0 \ne 0$.

**Requested Changes:**

1. Before I'd recommend acceptance, if my issues raised above are indeed valid, I would ask the authors to address two correctness issues. First, Proposition 4.2 and the downstream results depending on it would need to be repaired or weakened. The current proof appears to use a fixed-$y$ conditional identity (whose null set may depend on $y$) at the random value $y=g(\xi_1)$. This allows the diagonal counterexample above, and undermines the "automatic PIC" step used in Corollary 4.5 and Theorem 2.10. Definition 2.7 and Theorem 2.10 should also perhaps explicitly handle the base case $X_0$, since the current conditional-independence requirement begins with $X_{n+1}$ and (I think) does not rule out examples like $X_0=Z$.

2. Second, Counterexample 2.13 should be corrected (perhaps by using a nonzero-mean $Y$).


Minor spelling and grammar:
- (p1) best know -> best known
- (appendix B) measuralbe -> measurable
- (throughout) comma after However
- (throughout) capitalise Borel

---

> ### Author Response · Authors · 2026-05-28
>
> Thank you for your careful review and the identification of this problem. Since the reviews and their responses seem to be visible to everyone you may have already seen in our response to reviewer vYVd that your objection to Proposition 4.2 is indeed valid. Unfortunately we currently do not see a way to retain any of the downstream results even with stronger assumptions as explained in the answer to reviewer vYVd. We will try to find a way to fix this, but cannot offer any timeline at the moment.

---

### Comment · Action_Editor_nCo8 · 2026-03-20
**Review Timeline Update**

Authors: I have found a panel of 3 reviewers for this paper, so review will start shortly.

Reviewers: Given the technicality of this paper, you do not need to adhere to the usual TMLR timeframe for review, which is quite tight. Ignore the automated email reminders. I've set ~ 2 months as a reasonable timeline to review this one. The authors are already aware that this review process will be slower than usual for TMLR. If you end up needing more time, please let me know.

Thanks all!

---

### Decision · Action_Editor_nCo8 · 2026-07-04

**Recommendation:** Reject

**Audience:**

Yes

**Audience Explanation:**

Although I think the topic of this paper is borderline for the TMLR community, ultimately I think readers would be interested in a foundational understanding in the limitations of naive methods for inference with stochastic processes.

**Claims And Evidence:**

No

**Claims Explanation:**

Main theoretical results in the work have been found to be false.

**Resubmission Of Major Revision:**

The authors may consider submitting a major revision at a later time.